# MULTI-MODALITY EXPANSION AND RETENTION FOR LLMS THROUGH PARAMETER MERGING AND DECOUPLING

## ABSTRACT

Extensive fine-tuning of the synthesis between multimodal encoders and Large Language Models (LLMs) on modality-specific data can expand the modalities that LLM can handle, leading to the formation of Multimodal Large Language Models (MLLMs). However, this paradigm to expanding modalities heavily relies on initiating fine-tuning from scratch with new multimodal data, which is both resource-intensive and inflexible. In this paper, we propose *MMER (Multi-modality Expansion and Retention)*, a novel *training-free* approach that reuses and composes existing MLLMs to facilitate effective multimodal expansion while retaining the original performance of each MLLM. In particular, MMER maintains the multimodal encoders of the MLLMs while merging their LLM parameters. By comparing the original LLM parameters with the merged ones, MMER can create binary masks that enable an approximate separation of the LLM parameters for each modality. This process allows the decoupled parameters to independently process modality-specific inputs, thereby reducing parameter conflicts and maintaining the fidelity of the original MLLMs. Additionally, MMER integrates strategies to prevent catastrophic forgetting by employing a similar approach to separately decouple the parameters fine-tuned on new tasks from the original parameters. Experiments on three multimodal tasks and fourteen dual-modal tasks show significant improvements over recent baselines, demonstrating that MMER can effectively expand multimodal capabilities of LLMs while retaining 99.6% of the original performance. Further experiments in both single-task and cross modalities multi-task scenarios reveal that MMER significantly mitigates catastrophic forgetting.

## 1 INTRODUCTION

Large Language Models (LLMs) (Kenton & Toutanova, 2019; Touvron et al., 2023; Wu et al., 2023b) have recently become a cornerstone in artificial intelligence due to their exceptional performance. Building upon the capabilities of LLMs, researchers (Li et al., 2023b; Liu et al., 2023; Bai et al., 2023) integrate encoders for additional modalities and utilize extensive modality-text data for alignment. These integrated systems are then fine-tuned to develop Multimodal Large Language Models (MLLMs), which excel at processing multimodal information. This paradigm has led to the successful creation of numerous MLLMs across various modalities (Wu et al., 2024; Jiang et al., 2023).

Most MLLMs are specialized in dual modalities, with examples including vision-oriented LLMs like LLaVA (Liu et al., 2023) and MiniGPT-4 (Zhu et al., 2024b), as well as video LLMs (Lin et al., 2023; Maaz et al., 2024) and audio LLMs (Chu et al., 2023; Deshmukh et al., 2023). Despite these advancements, there is a growing impetus to expand the number of modalities that MLLMs can handle in order to address diverse applications. A straightforward method involves adding multiple new modality encoders or employing a unified multimodal encoder, followed by re-fine-tuning the MLLMs with fresh modality-text data. For instance, X-LLMs (Chen et al., 2023a) and MACAW-LLMs (Lyu et al., 2023) integrate encoders for images, videos, and audio with an LLM simultaneously, while OneLLM (Han et al., 2024) connects a universal encoder that handles eight distinct modalities with an LLM. These different architectures are then aligned and fine-tuned using new multimodal instruction data (e.g., image-speech chat). However, this kind of method lacks flexibility because it requires fine-tuning from scratch on new modality-text data to expand the multimodal capabilities

Figure 1: Illustration of the key idea of our MMER approach. Multi-Modality Expansion creates a versatile model from existing MLLMs through a **training-free and extensible** process. Multi-Modality Retention reconstructs the original MLLMs or new task MLLMs to retain their performance and mitigate catastrophic forgetting.

of MLLMs. Moreover, it necessitates the generation or acquisition of high-quality multimodal instruction data (Zhao et al., 2023), consuming substantial computational and human resources.

To overcome the aforementioned limitations, researchers have explored model merging or composition (Chen et al., 2024; Shukor et al., 2023; Panagopoulou et al., 2023) to facilitate multimodal expansion in MLLMs. For instance, Chen et al. (2024) proposed NaiveMC, a basic, training-free framework that reuses and combines modality-specific encoders from multiple MLLMs into the merged LLM, enabling it to handle multiple modalities. They further introduced the DAMC framework (Chen et al., 2024), which retrains existing MLLMs by separating modality parameters from language model parameters. This approach mitigates parameter interference in the merged MLLM, thereby enhancing the performance of multimodal expansion. However, these two frameworks encounter a trade-off: NaiveMC is train-free but delivers lower performance, whereas DAMC, though requiring training, yields better results. Nevertheless, both frameworks share a common drawback: they struggle to retain the original modality performance of the MLLMs due to parameter interference.

In this paper, we propose a novel training-free approach named MMER (Multi-modality Expansion and Retention), which achieves multimodal expansion while bypassing the trade-off of previous methods and retains the performance of each original MLLM (See Figure 1). First, we merge the *task vectors* (Ilharco et al., 2023), which represent the difference between the fine-tuned parameters and the pre-train LLM parameters, into a single merged task vector. Next, by comparing the Directional Congruence and Dominant Significance of the merged task vector with the original task vectors, we construct modality-specific binary masks. These masks effectively identify the modality-specific information retained in the merged task vector and approximately decouple it back into the original modality parameters without requiring additional training. This decoupling strategy enables the merged MLLM to independently process non-textual modality information using its corresponding reconstructed parameters, thereby significantly reducing interference from other modalities.

Additionally, by re-adding the decoupled modality task vectors back to the base LLM parameters and integrating the corresponding encoders, we are able to reconstruct the original MLLMs approximately. This strategy not only retains the performance of the original modalities but also saves storage space. Remarkably, our MMER approach exhibits strong resistance against catastrophic forgetting (Yang et al., 2023; Goodfellow et al., 2013) when handling new tasks. It can enhance performance on new tasks without diminishing effectiveness on previous tasks. The process begins with selecting the corresponding original MLLM based on the modality of the new task for fine-tuning. Then, the task vector of the newly fine-tuned MLLM is merged with the task vectors of all original MLLMs to create a merged task vector, and a corresponding mask is generated. This strategy effectively separates the parameters of the new task from the original parameters, thereby preventing damage to the original parameters during fine-tuning and significantly mitigating catastrophic forgetting.

We demonstrated the effectiveness of our MMER approach by composing four MLLMs (i.e., image, audio, video, and point cloud) and conducted extensive experiments across three scenarios. Firstly, in benchmarks like MCUB (Chen et al., 2024), which involve more than two modalities, MMER achieves a significant improvement compared to NaiveMC framework and various model merging methods such as TA (Ilharco et al., 2023) and TIES (Yadav et al., 2023). This confirms that MMER realizes and enhances the multimodal expansion capabilities of LLMs without requiring additional

training. Secondly, we examined the performance of MLLMs approximately reconstructed by MMER on fourteen dual-modal benchmarks covering four modalities. The experiments reveal that these MLLMs fully retain their original performance. Lastly, we verified MMER's resistance to catastrophic forgetting in both single-task and cross modalities multi-task scenarios. Our results show that MMER retained 98% of its performance on previous tasks while efficiently adapting to new tasks.

In summary, our work makes several significant **contributions**: (i) We propose a training-free MMER approach that enables seamless multimodal expansion for LLMs through multimodal parameter merging and decoupling. (ii) We demonstrate two additional practical applications of the MMER approach: retaining the performance of original MLLMs and mitigating catastrophic forgetting in MLLMs. (iii) We conduct extensive and rigorous experiments on various multimodal benchmarks across three scenarios, and the experimental results confirm the effectiveness of the MMER approach.

## 2 RELATED WORK

### 2.1 MULTIMODAL LARGE LANGUAGE MODELS

As text-based LLMs are insufficient to meet evolving demands, significant research (Dai et al., 2023; Li et al., 2023a; Achiam et al., 2023; Ye et al., 2023) efforts are focused on developing LLMs that can effectively handle multimodal inputs. Vision LLMs (Alayrac et al., 2022; Li et al., 2023b; Liu et al., 2023), as trailblazers in this field, have excelled across numerous vision-language tasks by connecting pre-trained visual encoders to LLMs through various alignment layers. Similarly, other modalities like audio (Rubenstein et al., 2023; Deshmukh et al., 2023) and video (Lin et al., 2023; Maaz et al., 2024) have rapidly followed suit, resulting in a surge of dual-modality MLLMs. Meanwhile, researchers have explored unifying multiple modalities into a single LLM. Approaches like Pandagpt (Su et al., 2023) and ImageBind-llm (Han et al., 2023) connect a unified multimodal encoder like ImageBind (Girdhar et al., 2023) with an LLM but rely only on image-text data, leading to suboptimal performance. OneLLM (Han et al., 2024) improves on this by introducing a versatile encoder for eight modalities and aligning all of them with language. However, these methods cannot expand modalities due to the encoders have fixed input modalities. Another approach connects multiple modality-specific encoders to an LLM, as seen in X-LLM (Chen et al., 2023a) and MACAW-LLM (Lyu et al., 2023), which integrate encoders for images, videos, and audio. AnyMAL (Moon et al., 2023) further adds an IMU encoder. These methods typically require high-quality multimodal data for joint training and still struggle with modality expansion. In contrast, our MMER approach provides an efficient, training-free solution for seamless multimodal expansion in LLMs.

### 2.2 MODEL MERGING AND MODEL COMPOSITION

The proliferation of various model checkpoints has sparked concerns over data privacy and resource efficiency, shifting focus toward model merging and composition (Yang et al., 2024; Lu et al., 2024). Model merging, which merges multiple models fine-tuned from the same initialization, can improve single-task performance (Gupta et al., 2020; Wortsman et al., 2022), out-of-distribution generalization (Arpit et al., 2022; Ramé et al., 2022), or combine their capabilities (Wan et al., 2024; Ilharco et al., 2022). For example, TA (Ilharco et al., 2023) defines the concept of task vectors and uses arithmetic operations like addition to merge models. TIES (Yadav et al., 2023) mitigates interference during merging by pruning redundant parameters and resolving sign conflicts, while DARE (Yu et al., 2024) proposes a preprocessing step that randomly drops and scales parameters to achieve the same goal. TALL-masks (Wang et al., 2024) proposes an algorithm to identify original task information from the merged task vector and eliminate harmful parameters to enhance merging performance. Model merging has been further applied to multimodal models (Aiello et al., 2024; Wu et al., 2023a). Sung et al. (2023) and Sundar et al. (2024) investigated merging multimodal transformers to improve performance in specific tasks, such as speech recognition. Model Tailor (Zhu et al., 2024c) merges the original model with a fine-tuned model to mitigate catastrophic forgetting in MLLMs. However, they do not explore the merging of multiple MLLMs. The NaiveMC and DAMC frameworks (Chen et al., 2024) address this by using model merging and composition to create a unified MLLM that inherits the modality capabilities of multiple MLLMs, thus enabling seamless expansion into new modalities. Nonetheless, each framework has its limitations: one necessitates additional training, while the other delivers subpar performance. In contrast, MMER approach avoids both of these drawbacks. It enhances the multimodal expansion capabilities of MLLMs

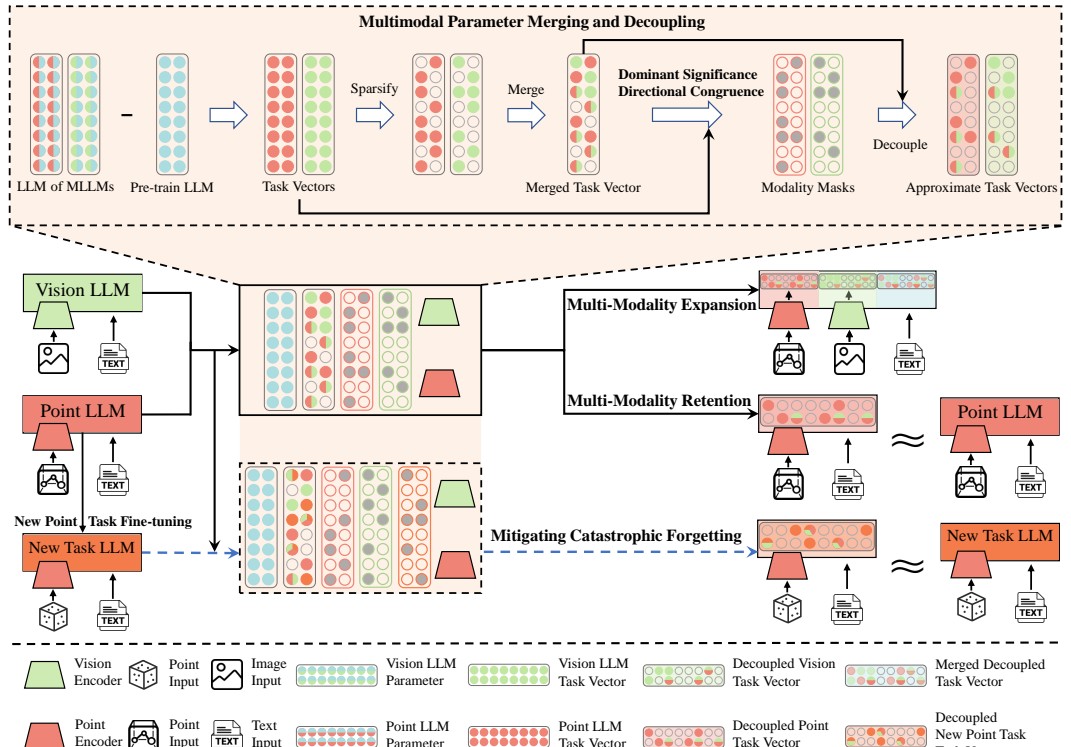

Figure 2: Illustration of the details of our MMER approach with only **Image** and **Point Cloud** modalities are considered **for clarity**. Each block corresponds to the same weight matrix, with empty blocks denoting zero value. The symbol "≈" signifies similar performance. We separate modality-specific components from the base fine-tuned LLMs, then apply parameter merging and decoupling to these LLMs to generate masks and merged task vector. Different task vectors are decoupled to reconstruct original MLLMs based on the specific scenario.

without requiring additional training, while nearly retaining the original modalities' performance. Additionally, it demonstrates impressive resistance to catastrophic forgetting when adapting to new tasks. Appendix A provides a detailed comparison of MMER approach with related methods.

## 3 METHODOLOGY

In our MMER approach, we first merge the LLM parameters $\{\theta_1, \theta_2, \ldots, \theta_n\}$ from multiple MLLMs, all fine-tuned from the same base LLM $\theta_{\text{pre}}$, into a unified LLM. However, such a merged model is particularly susceptible to interference between parameters from different modalities, which can degrade the performance of modality-specific representations. To overcome this challenge, we adopt a training-free parameter decoupling method to effectively enhance the multimodal performance of the merged LLM while also ensuring that the performance of each original modality is retained. This method approximately decouples modality-specific parameters within the merged parameter, ensuring that the representation of each non-text modality is processed independently by their respective parameters. A visual workflow illustrating our MMER approach is depicted in Figure 2.

### 3.1 MULTIMODAL PARAMETER MERGING AND DECOUPLING

The key idea of the MMER approach is its mechanism of the training-free multimodal parameter merging and decoupling. Specifically, we commence by employing the advanced model merging technique Ties (Yadav et al., 2023) to merge $\{\theta_1, \theta_2, \ldots, \theta_n\}$. Ties first calculates the task vectors for each MLLM as $\tau_{i,pre} = \theta_i - \theta_{\text{pre}}$. Subsequently, it refines these task vectors $\tau_{i,pre}$ by selecting the Top $K\%$ extreme values to filter out non-essential parameters, which yields sparse task vectors $\tau_i$. These sparse task vectors are then merged according to sign consistency to generate the merged task vector $\tau_* = \text{merge}(\sum_{i=1}^{n} \tau_i)$ and the final merged LLM parameter is $\theta_* = \theta_{\text{pre}} + \alpha \cdot \tau_*$, where $\alpha > 0$

is a scaling factor determined by the accuracy of the validation dataset. This dataset is constructed by selecting one task from each modality and combining their respective validation sets.

Previous studies (Panigrahi et al., 2023; Wang et al., 2024) have shown that after merging multiple different task vectors, the majority of their parameter information is retained and embedded into the resulting merged task vector $\tau_*$. By comparing the original MLLM task vectors $\tau_i$ with the merged task vector $\tau_*$, we can identify the relevant parameter subsets specific to each original modality task vector. This enables the construction of modality-specific binary masks $m_i$, which allow decoupling and approximating of the parameters for the original modality task vectors $m_i \circ \tau_*$. These binary masks filter out irrelevant parameters in the merged task vector, preserving only modality-specific information and reconstructing the original model parameters $\hat{\theta}_i$:

$$\hat{\theta}_i = \theta_{\text{pre}} + m_i \circ \tau_* \approx \theta_i \tag{1}$$

We construct the binary masks $m_i$ by minimizing the Manhattan distance $\ell_1^*$ between the reconstructed model parameters $\hat{\theta}_i$ and the LLM parameters $\theta_i$ of original MLLMs:

$$\underset{m_i \in \{0,1\}^P}{\arg\min} \left| \hat{\theta}_i - \theta_i \right| = \underset{m_i \in \{0,1\}^P}{\arg\min} |m_i \circ \tau_* - \tau_i| = \underset{m_i \in \{0,1\}^P}{\arg\min} \sum_{p=1}^{P} \left| m_i^{(p)} \circ \tau_*^{(p)} - \tau_i^{(p)} \right| \tag{2}$$

where $P$ represents the total number of parameters. If the sign of $\tau_i^{(p)}$ is inconsistent with that of $\tau_*^{(p)}$, i.e., $\text{sign}(\tau_i^{(p)}) \neq \text{sign}(\tau_*^{(p)})$, the binary masks $m_i^{(p)}$ is set to 0 to avoid directional conflict. This step is referred to as **Directional Congruence**. Conversely, when the sign of $\tau_i^{(p)}$ aligns with $\tau_*^{(p)}$ and $\left| \tau_i^{(p)} \right| \geq \left| \tau_*^{(p)} - \tau_i^{(p)} \right|$, i.e., $\left| \tau_i^{(p)} \right| \geq 50\% \left| \tau_*^{(p)} \right|$, this indicates that $\tau_i^{(p)}$ is a dominant component of the merged parameter $\tau_*^{(p)}$. Thus, $\tau_*^{(p)}$ can be approximated as $\tau_i^{(p)}$, and $m_i^{(p)}$ is set to 1, which we refer to as **Dominant Significance**. Additionally, we introduce a scaling factor $\lambda_i$ to refine the selection process of Dominant Significance to accommodate the varying numbers and modalities of original MLLMs. A smaller $\lambda_i$ results in the selection of more parameters. The choice of $\lambda_i$ is guided by validation dataset accuracy, enabling different masks $m_i$ to be assigned with different values of $\lambda_i$. The final mask $m_i$ is constructed using the following formula:

$$m_i = \begin{cases} 1 & \text{if } |\tau_i^{(p)}| \geq \lambda_i \cdot 50\%|\tau_*^{(p)}| \text{ and } \text{sign}(\tau_i^{(p)}) = \text{sign}(\tau_*^{(p)}) \\ 0 & \text{otherwise} \end{cases} \tag{3}$$

Our subsequent experiments validate the effectiveness of this parameter decoupling strategy. As illustrated in Figure 4, by employing binary masks to separate the pertinent subset of critical parameters, the original parameters can be approximately decoupled, thereby retaining the original performance.

## 3.2 The MMER Approach

We now present a comprehensive exposition of how the aforementioned multimodal parameter merging and decoupling method not only facilitates both the multi-modality expansion and retention but also addresses the challenge of catastrophic forgetting in MLLMs.

### 3.2.1 Multi-Modality Expansion

Typical MLLMs are composed of two key elements: modality-specific components, including modality encoders and alignment layers, and a base fine-tuned LLM. Our MMER approach begins by disentangling these components. MMER proceeds by applying the multimodal parameter merging and decoupling strategy to these base fine-tuned LLMs of multiple MLLMs, yielding a merged task vector $\tau_*$, the pre-trained LLM parameter $\theta_{\text{pre}}$, and $n$ modality-specific binary masks $m_i$. The corresponding modal-specific components, including their weights, are retained and reutilized directly, enabling the merged MLLM to seamlessly process all original modalities without loss of functionality.

As depicted in Figure 3, upon receiving multimodal data, MMER initially identifies the pertinent modality-specific components according to the data type and respectively encodes them into representation inputs $X = [X_{M_1}, \ldots, X_{M_n}, X_t]$, where $X_{M_i}$ and $X_t$ represent the modality-specific sequences and text sequences. MMER then dynamically decouples the approximate modality-specific

parameters $\theta_{\text{pre}} + m_i \circ \tau_*$, adeptly separating the parameters for each modality. This crucial step ensures that non-text modality representations are processed independently with their corresponding modality parameters. It is noteworthy that text representations are processed with the merged parameter $\theta_{\text{pre}} + \overline{m} \circ \tau_*$, where $\overline{m}$ represents the average of all modality-specific masks $m_i$. For example, when the representations progress to the attention mechanism at the $l$-th layer, MMER first decouples the modality-specific parameter weights $W_{*,l}^Q$ of the queries weights in the $l$-th layer from $\tau_*$, before proceeding with the subsequent computation:

$$\mathbf{Q}_l = \left[ X_{M_1,l}\left( m_{1,l}^Q \circ W_{*,l}^Q + W_{pre,l}^Q \right), \ldots, X_{t,l}\left( \overline{m}_l^Q \circ W_{*,l}^Q + W_{pre,l}^Q \right) \right] \tag{4}$$

where $W_{pre,l}^Q$ represents the weights of the queries in the $l$-th layer form $\theta_{\text{pre}}$. Subsequently, MMER sequentially decouples the modality-specific parameters for the keys and values in the $l$-th layer, and executes the computations to derive $\mathbf{K}_l$ and $\mathbf{V}_l$. Following this, we carry out the attention operation:

$$X_l^a = Attention(\mathbf{Q}_l, \mathbf{K}_l, \mathbf{V}_l) \tag{5}$$

$$[X_{M_1,l}^a, \ldots, X_{M_n,l}^a, X_{t,l}^a] = Split(X_l^a) \tag{6}$$

Please note that the output representation should be partitioned according to the different modalities to match the input form. Consequently, the final output of the attention mechanism at the $l$-th layer is:

$$[X_{M1,l}^o, \ldots, X_{t,l}^o] = \left[ X_{M_1,l}^a\left( m_{1,l}^O \circ W_{*,l}^O + W_{pre,l}^O \right), \ldots, X_{t,l}^a\left( \overline{m}_l^O \circ W_{*,l}^O + W_{pre,l}^O \right) \right] \tag{7}$$

Such an implementation effectively alleviates parameter conflicts across different modalities, thereby ensuring that the merged MLLM can maintain relative fidelity when processing multimodal data.

### 3.2.2 Multi-modality Retention

Previous studies, such as model merging (Ilharco et al., 2023; Yadav et al., 2023) and NaiveMC (Chen et al., 2024), have demonstrated that performance tends to degrade (See Table 2) when handling the original tasks, primarily due to the discrepancies between the merged model parameter and the original model parameters. However, MMER effectively circumvents this issue, ensuring the original MLLMs' performance is retained. When dealing with tasks that are specific to the modalities of the original MLLMs, MMER can directly reconstruct an approximate version of the original MLLMs to perform inference on these tasks. The procedure is as follows: First, decouple the modality-specific task vector $m_i \circ \tau_*$ from the merged task vector, then add it to the pretrained LLM parameter $\theta_{\text{pre}}$ to obtain the restored LLM $\hat{\theta}_i = \theta_{\text{pre}} + m_i \circ \tau_*$. Finally, integrate the corresponding modality-specific components with the restored LLM to construct the final reconstructed MLLM. This strategy effectively mitigates interference from parameters of other MLLMs and retains the original performance of each MLLM. Subsequent experiments, as illustrated in Table 2 validate the effectiveness of this strategy.

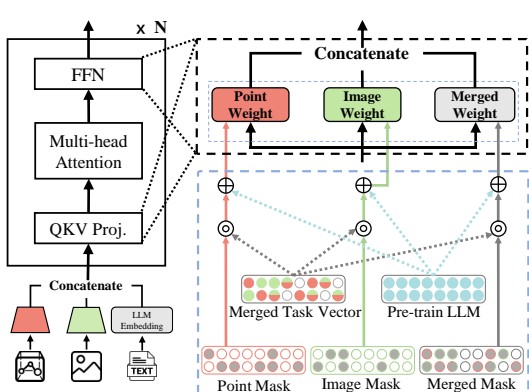

Figure 3: Details of dynamic parameter decoupling in MMER for handling multimodal data. $\odot$ and $\oplus$ represent the Hadamard product and addition operation.

### 3.2.3 Mitigating Catastrophic Forgetting

Typically, fine-tuning MLLMs on new data enhances their performance on newly introduced tasks, but it often comes at the cost of diminished performance on the previous tasks (Goodfellow et al., 2013). This degradation in performance, referred to as catastrophic forgetting, arises when fine-tuning for new tasks. Drawing on the insight that most parameter information from individual task vectors is retained in the merged task vector, we can leverage MMER to additionally mitigate catastrophic forgetting. Specifically, we first select the corresponding original MLLM based on the modality of the new task and proceed to fine-tune its LLM using the new data, while keeping the modality-specific components frozen. Next, we apply the parameter merging and decoupling method to the fine-tuned MLLM, alongside all original MLLMs, generating a new merged task vector and modality- or task-specific binary masks. For the new task, we apply the task-specific mask to recover the fine-tuned MLLM for processing. For the previous tasks, we maintain the use of the previously proposed multi-modality expansion or retention strategy. This method enables MMER to effectively adapt to new tasks while retaining its performance on previous tasks, thereby substantially mitigating the risk of catastrophic forgetting.

Table 1: Accuracy (%) results on multimodal benchmarks with different combinations of video (V), image (I), audio (A), point cloud (P), and text (T) inputs. MCUB and MUSIC-AVQA consist of five and three subsets, respectively. TA and TIES serve as alternatives to the average merging method utilized in NaiveMC framework. Optimal results are highlighted in boldface, while sub-optimal results are underlined.

| Task (→) | ModelNet40 | MUSCI-AVQA | | | MCUB | | | | | Avg |
| Method (↓) | PI-T | IA-T | VI-T | VA-T | AVI-T | AVP-T | AIP-T | VIP-T | AVIP-T | |
|---|---|---|---|---|---|---|---|---|---|---|
| NaiveMC[ACL24] (Chen et al., 2024) | 60.53 | 39.31 | 47.65 | 47.40 | 53.64 | 56.28 | 60.53 | 54.60 | 59.16 | 53.23 |
| TA[ICLR23] (Ilharco et al., 2023) | 62.04 | 40.22 | 47.97 | 46.70 | 53.44 | 56.28 | 63.36 | 55.40 | 59.72 | 53.90 |
| TIES[NeurIPS23] (Yadav et al., 2023) | 61.74 | 43.27 | 49.27 | 48.60 | 53.64 | 55.47 | 61.74 | 54.60 | 58.55 | 54.10 |
| NaiveMC (w/ DARE[ICML24] (Yu et al., 2024)) | 60.32 | 39.78 | 47.98 | 47.67 | 53.64 | 56.68 | 60.73 | 54.80 | 59.53 | 53.46 |
| TA (w/ DARE) | **62.75** | 40.46 | 47.98 | 46.92 | 54.25 | 56.48 | 64.17 | 55.40 | 60.08 | 54.27 |
| TIES (w/ DARE) | 61.96 | 43.78 | 49.54 | 48.98 | 54.25 | 55.87 | 62.55 | 55.20 | 59.06 | 54.57 |
| **MMER (ours)** | 62.15 | **47.25** | **51.27** | **51.77** | **56.48** | **59.31** | **65.59** | **56.00** | **61.63** | **56.82** |

## 4 EXPERIMENTS SETUP

### 4.1 IMPLEMENTATION

In our forthcoming experiments, we explored our MMER approach across four MLLMs: Image, Audio, Video, and Point Cloud LLMs, all built upon Vicuna-7B-v1.5 (Zheng et al., 2023). To ensure fairness and comparability in our analysis, we fine-tuned these four MLLMs in the same environment, following previous works (Liu et al., 2024; Lin et al., 2023; Xu et al., 2024; Panagopoulou et al., 2023). Details regarding the fine-tuning of these MLLMs are provided in Appendix B.2. For the hyperparameters in the parameter merging and decoupling process, we set $\alpha$ to 1 and $K$ to 80%, while $\lambda$ was calibrated according to the different modalities.

### 4.2 BASELINE METHODS

In the multi-modality expansion and retention experiments, we first compared MMER with the training-free modal composition framework NaiveMC (Chen et al., 2024). We excluded the comparison with DAMC framework (Chen et al., 2024), as its reliance on additional training introduces variables that could compromise the fairness of our experimental evaluation. Additionally, since model merging methods can substitute the average merging strategy in NaiveMC framework to enhance multimodal performance, we also compared against training-free model merging baselines, including TA (Ilharco et al., 2023) and TIES (Yadav et al., 2023). Given that DARE (Yu et al., 2024) is complementary to other model merging methods, we combined it with TA, TIES, and NaiveMC framework in our experiments. Additionally, we included the zero-shot performance of the original MLLMs as a performance upper bound. We evaluated the performance based on both the evaluation scores or accuracy and the performance retention, the latter of which is defined in Appendix B.1.

### 4.3 DATASETS AND BENCHMARKS

In the multi-modality expansion experiments, we evaluated several multimodal benchmarks containing more than two modalities, including MCUB (Chen et al., 2024), MUSIC-AVQA (Li et al., 2022), and ModelNet40 (Wu et al., 2015) with images. Furthermore, we assessed the multi-modality retention across fourteen dual-modal benchmarks spanning four modalities: image, video, audio, and point cloud. Specifically, the image tasks include VQAv2 (Goyal et al., 2017), GQA (Hudson & Manning, 2019), TextVQA (Singh et al., 2019), VizWiz (Gurari et al., 2018), ScienceQA (Lu et al., 2022), POPE (Li et al., 2023c), and OK-VQA (Marino et al., 2019). The audio tasks cover TUT (Mesaros et al., 2017), VocalSound (Gong et al., 2022), and Clotho (Drossos et al., 2020). The video benchmarks include MSRVTT (Xu et al., 2016) and MSVD (Chen & Dolan, 2011), while the point cloud tasks focus on ModelNet40 (Wu et al., 2015) and Objaverse (Deitke et al., 2023). Finally, we evaluated MMER's resilience to catastrophic forgetting in both single-task and cross modalities multi-task settings across two new tasks, Flickr30k (Young et al., 2014) and Clotho-AQA (Lipping et al., 2022).

## 5 MAIN RESULTS

**Multi-Modality Expansion.** Table 1 summarizes the primary results of the multi-modality expansion experiments across four benchmarks. The notation "XY-T" denotes the combination of modalities included in the dataset, where A stands for Audio, P for Point Cloud, V for Video, I for Image, and T for Text. For example, "IA-T" signifies that the dataset comprises Image, Audio, and Text modalities. Regardless of the input data modalities, we utilized the unified MLLM, derived from merging MLLMs across four distinct modalities, to process the data. We can draw the following observations: **(i)** Various advanced model merging methods improve the performance of the NaiveMC framework, suggesting that training-free model merging methods can still be effectively applied to the merging of MLLMs. This is a domain that has not been previously explored. Additionally, this implies that there are considerable parameter conflicts in the merged MLLM, as

Table 2: Results on fourteen dual-modal benchmarks spanning four modalities. The performance retention from the original MLLMs are shown in parentheses. Original MLLMs refer to their zero-shot results. "Trimmed Avg" represents the average result obtained after excluding three point or audio classification tasks.

| Task (→) | 2 Point Tasks | 3 Audio Tasks | 2 Video Tasks | 7 Image Tasks | Trimmed Avg |
|---|---|---|---|---|---|
| Method (↓) | Score (%) / Acc.(%) | Score (%) / Acc.(%) | Acc.(%) | Acc.(%) | Score (%) / Acc.(%) |
| Original MLLMs | 23.15 / 21.27 | 25.30 / 24.71 | 39.79 | 62.23 | 24.23 / 51.01 |
| NaiveMC [ACL2024] (Chen et al., 2024) | 22.65 $_{(97.8)}$ / 20.49 $_{(96.3)}$ | 24.59 $_{(97.2)}$ / 30.65 $_{(124.8)}$ | 36.92 $_{(93.0)}$ | 52.56 $_{(83.6)}$ | 23.62 $_{(97.5)}$ / 44.59 $_{(88.3)}$ |
| TA [ICLR23] (Ilharco et al., 2023) | 22.96 $_{(99.2)}$ / 21.02 $_{(98.8)}$ | 24.68 $_{(97.5)}$ / 31.88 $_{(129.8)}$ | 37.57 $_{(94.5)}$ | 54.89 $_{(87.5)}$ | 23.82 $_{(98.3)}$ / 46.23 $_{(91.0)}$ |
| TIES [NeurIPS23] (Yadav et al., 2023) | 22.82 $_{(98.6)}$ / 20.83 $_{(97.9)}$ | 24.79 $_{(98.0)}$ / 32.15 $_{(130.9)}$ | 37.81 $_{(95.1)}$ | 54.10 $_{(86.2)}$ | 23.80 $_{(98.3)}$ / 45.96 $_{(90.6)}$ |
| NaiveMC (w/ DARE[ICML2024] (Yu et al., 2024)) | 22.83 $_{(98.6)}$ / 20.77 $_{(97.6)}$ | 24.72 $_{(97.7)}$ / 31.62 $_{(128.8)}$ | 37.63 $_{(94.4)}$ | 53.61 $_{(85.3)}$ | 23.78 $_{(98.1)}$ / 45.62 $_{(89.8)}$ |
| TA (w/ DARE) | 23.04 $_{(99.5)}$ / 21.25 $_{(99.9)}$ | 24.82 $_{(98.1)}$ / 32.44 $_{(132.0)}$ | 37.52 $_{(94.4)}$ | 55.47 $_{(88.4)}$ | 23.95 $_{(98.8)}$ / 46.50 $_{(91.4)}$ |
| TIES (w/ DARE) | 22.76 $_{(98.3)}$ / 20.98 $_{(98.6)}$ | 24.92 $_{(98.5)}$ / 33.02 $_{(134.4)}$ | 38.00 $_{(95.6)}$ | 54.73 $_{(87.2)}$ | 23.84 $_{(98.4)}$ / 46.37 $_{(91.4)}$ |
| **MMER (ours)** | **23.14 $_{(99.9)}$ / 22.49 $_{(105.7)}$** | **25.20 $_{(99.6)}$ / 38.51 $_{(155.6)}$** | **39.28 $_{(98.5)}$** | **62.40 $_{(100.3)}$** | **24.17 $_{(99.8)}$ / 50.84 $_{(99.4)}$** |

Table 3: Results on previous and new tasks in the single-task scenario and **cross-modalities** multi-task scenario. MMER-xx refers to merging the MLLM fine-tuned on a new task xx into MMER. MMER-Clotho-AQA+Flickr30k denotes the merging of both the audio LLM fine-tuned on Clotho-AQA and the vision LLM fine-tuned on Flickr30k into MMER. The symbol "(∼)" signifies performance retention.

| Task (→) | Previous Tasks | | | | | New Tasks | |
|---|---|---|---|---|---|---|---|
| | 2 Point tasks | 3 Audio tasks | 2 Video tasks | 7 Image tasks | 3 Multimodal tasks | Clotho-AQA | Flickr30k |
| Baseline (↓) | Score / Acc. | Score / Acc. | Acc. | Acc. | Acc. | Acc. | Score |
| Original MLLMs | 23.15 / 21.27 | 25.30 / 24.71 | 39.79 | 62.23 | - | 49.40 | 51.26 |
| Fine-tune on Clotho-AQA | - | 19.82 / 12.31 (↓) | - | - | - | 57.80 (↑) | - |
| Fine-tune on Flickr30k | - | - | - | 57.25 (↓) | - | - | 57.71 (↑) |
| MMER | 23.14 / 22.49 | 25.20 / 38.51 | 39.28 | 62.40 | 56.82 | 49.28 | 51.00 |
| **MMER-Clotho-AQA** | 22.95 / 21.87 | 25.12 / 38.23 (∼) | 39.17 | 62.20 | 56.53 | 57.71 (↑) | 50.94 |
| **MMER-Flickr30k** | 23.05 / 22.03 | 24.96 / 37.68 | 38.90 | 62.27 (∼) | 56.44 | 48.94 | 57.08 (↑) |
| **MMER-Clotho-AQA+Flickr30k** | 22.82 / 21.56 | 24.88 / 37.69 (∼) | 38.53 | 61.94 (∼) | 55.89 | 57.52 (↑) | 56.72 (↑) |

these methods primarily focus on mitigating conflicts among merging parameters. **(ii)** Our proposed MMER approach significantly outperforms NaiveMC across all input combinations and tasks. This demonstrates that MMER effectively extends multimodal capabilities and enhances the merged MLLMs' ability to manage these modality combinations without requiring additional training for each specific combination. **(iii)** Furthermore, MMER achieves superior performance on all tasks, except for ModelNet40, when compared with model merging methods. This indicates that directly decoupling parameters after merging is more effective than merely reducing parameter conflicts during the merging process. **(iv)** We observe that as the number of merging modalities increases, the relative accuracy improvement of TIES (w/ DARE) compared to NaiveMC diminishes (i.e., from 4.81% to 0.94% to -0.17%), indicating that mutual interference among merging parameters intensifies. However, although the relative accuracy improvement of MMER declines, it remains significant (i.e., from 9.00% to 5.21% to 4.17%), indicating that parameter decoupling remains effective despite severe parameter conflicts.

**Multi-Modality Retention.** To evaluate how effectively each method retains the performance of the original MLLMs, we conducted experiments across tasks corresponding to the original MLLMs. The results are recorded in Table 2, where we observe the following: **(i)** Interestingly, all methods demonstrate substantial performance improvements on specific audio and point cloud tasks, which we attribute to task-specific differences. Specifically, these tasks, TUT, VocalSound, and ModelNet40, are all classification tasks, whereas the others involve captioning or question-answering tasks. The original audio and point cloud LLMs are not fine-tuned for classification tasks, resulting in their failure to follow instructions and leading to poorer zero-shot performance in these tasks. However, parameter merging unlocks their ability to follow instructions, as the training data for the vision LLM included similar instructions. This resulted in performance improvements even when the merged parameter is decoupled. To account for the impact of these tasks, we additionally provide the average performance trimming these tasks in Table 2 for a more accurate comparison of the various methods. **(ii)** Although NaiveMC successfully achieves multimodal expansion for handling multimodal tasks, its performance on the original tasks shows a substantial deviation from that of the original MLLMs. While varied model merging methods can partially alleviate the decline in original performance, the gap remains significant. In contrast, our MMER almost completely retains the original performance. For instance, in the trimmed average performance, our method attains 99.8% and 99.4% performance retention of the original MLLMs' evaluation scores and accuracy, respectively. Detailed performance for each task is provided in Appendix D.2.

**Mitigating Catastrophic Forgetting.** Finally, Table 3 illustrates that MMER effectively alleviates catastrophic forgetting, whether in single-task or multi-task scenarios across different modalities. **(i)** Clotho-AQA represents an audio task, while Flickr30k represents a visual task. Thus, we fine-tuned the original audio LLM and vision LLM separately on the respective new task. We observe that while fine-tuning MLLMs improves performance on new tasks, it tends to degrade performance on previous tasks in the corresponding modalities. In contrast, the MMER approach, which additionally incorporates a fine-tuned MLLM (i.e., MMER-Clotho-AQA and MMER-Flickr30k), demonstrates strong robustness. It maintains nearly the original performance across various previous tasks and adapts effectively to new tasks, achieving results comparable to those of fine-tuned MLLMs.

Table 4: Performance retention & Storage vs. Mitigating MLLMs' catastrophic forgetting methods in the **same modality**. Let $N$, $P$, and $P'$ represent the number of new tasks, the total LLM parameters, and the modality-specific component parameters, assuming each float parameter occupies 32 bits.

| Method | One New Task | | Two New Tasks | | Storage |
|---|---|---|---|---|---|
| | Previous tasks | New task | Previous tasks | New tasks | |
| Model Tailor[ICML24] (Zhu et al., 2024c) | 96.47 % | 91.69 % | 99.28 % | 87.50 % | $32(P + P')$ |
| **MMER (ours)** | **99.86 %** | **99.67 %** | **99.63 %** | **99.42 %** | $64P + 32P' + NP$ |

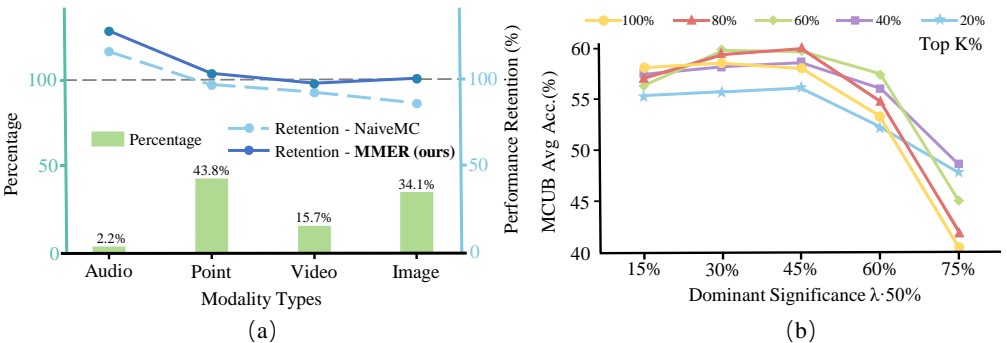

(a)                                                     (b)

Figure 4: **(a).** The bar plot illustrates the percentage of parameters selected by different modality masks, while the lines depict the performance retention of NaiveMC and the MLLMs reconstructed by MMER across various dual-modal benchmarks. **(b).** The line plots depict the variations in MCUB average accuracy across different merging sparsity ratios (Top $K\%$) and Dominant Significance ($\lambda \cdot 50\%$).

**(ii)** Additionally, we integrated both fine-tuned MLLMs into the MMER approach together to showcase its performance in cross modalities multi-task scenario. As the number of integrating MLLMs increases, MMER continues to maintain performance across various new and previous tasks to some extent. However, its ability to preserve performance does slightly diminish. Detailed performance for each task is provided in Appendix D.2.

In Table 4, we compare our MMER approach with the latest method for mitigating catastrophic forgetting in MLLMs within the same modality, since Model Tailor (Zhu et al., 2024c) is unable to accommodate new tasks across different modalities. The results show that MMER consistently outperforms Model Tailor in both single-task and multi-task scenarios, highlighting its effectiveness. Furthermore, as the number of new tasks increases, MMER maintains relatively stable performance, whereas Model Tailor exhibits a significant decline in performance on new tasks (i.e., from 91.69% to 87.50%), despite some improvement on previous tasks. However, a minor drawback of MMER is that its storage cost is approximately twice that of Model Tailor. Nonetheless, as the number of new tasks grows, MMER's practicality becomes more pronounced, making it a more viable solution in scenarios where balancing performance and storage efficiency is crucial.

## 6  ADDITIONAL RESULTS AND ANALYSIS

**Modality-Specific Masks Analysis.** Figure 4 (a) illustrates the percentage of parameters selected by different modality masks and compares the performance retention of MLLMs reconstructed with MMER against NaiveMC. It can be observed that MMER outperforms NaiveMC across all four modalities, with performance that is close to or even exceeds the original levels. This indicates that all integrated knowledge is preserved after merging, and the parameter subsets selected by the masks retain crucial modality-specific information. Additionally, we find that the audio mask, despite retaining only 2.2% of the parameters, still contributes to performance improvement. This finding is consistent with previous research (Yu et al., 2024), which noted that "Supervised fine-tuned language models tend to acquire excessively redundant delta parameters (also known as task vectors)." Our results further confirm that this observation holds true for MLLMs as well.

**Hyperparameters Analysis.** Figure 4 (b) explores the impact of the hyperparameters Top $K\%$ and the scaling factor $\lambda$ in multimodal parameter merging and decoupling. Firstly, Top $K\%$ controls the sparsity of the original task vectors. When sparsity is excessively high (e.g., 20% or 40%), performance deteriorates markedly due to the insufficient information retained in the sparse parameters. Conversely, in the absence of sparsity (e.g., 100%), the method fails to mitigate parameter conflicts among the original parameters, thereby hindering the approximate decoupling of modality parameters. The effect of the scaling factor $\lambda$ is akin to that of Top $K\%$. Specifically, the scaling factor $\lambda$ regulates the extent of information that the mask extracts from the merged task vector. If $\lambda$ is set too high, the decoupled modality parameters will contain insufficient effective information, leading to performance collapse (e.g., 60% or 75%). Conversely, if $\lambda$ is too low (e.g., 15%), irrelevant parameters remain, resulting in poor performance. In summary, Top $K\%$ and the scaling factor $\lambda$ work in tandem to regulate the amount of effective information contained in the decoupled parameters. For instance, the performance with $K$ set to 60% and $\lambda$ to 30% closely mirrors the results when $K$ is set to 80% and $\lambda$ to 45%.

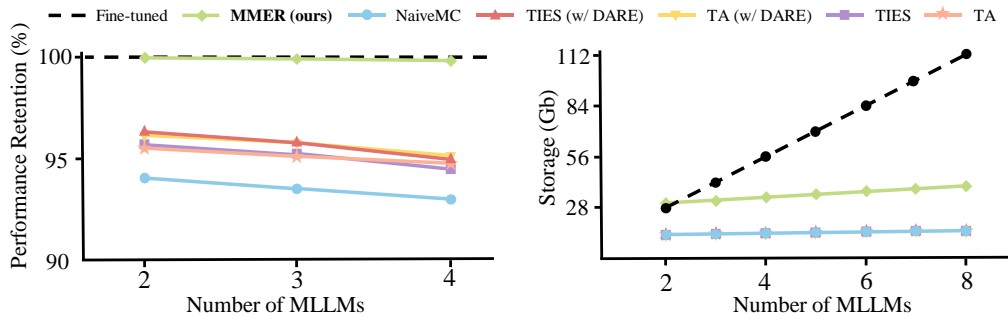

Figure 5: Performance retention and Storage cost vs. Number of original MLLMs in multi-modality retention experiments. Our MMER approach can nearly retain the initial performance across various combinations of MLLMs while significantly compressing the fine-tuned checkpoints.

**Performance & Storage vs. MLLM Quantity.** Finally, Figure 5 presents the performance retention and storage costs of merging varying numbers of MLLMs in multi-modality retention experiments. Our observations suggest that as the number of merging MLLMs increases, performance declines across all methods. This indicates that as more MLLMs are merged, conflicts between the merging parameters intensify. Nevertheless, our MMER consistently outperforms model merging methods, experiencing only minor performance degradation. In contrast, model merging methods exhibit a noticeable performance drop when dealing with multiple MLLMs. This highlights that parameter decoupling is robust in effectively mitigating parameter conflicts.

Although model composition or merging methods maintain low storage costs that remain constant regardless of the number of merging MLLMs, their lower performance may constrain their practical applicability. In contrast, maintaining individual MLLMs preserves strong performance for their respective modalities but fails to achieve multimodal expansion and results in linear growth in storage costs. Our MMER approach strikes an effective balance between these approaches. It enables multimodal expansion while retaining nearly 100% of the original MLLMs' modality capabilities and provides additional resilience against catastrophic forgetting. Moreover, it significantly reduces storage costs, as shown in Figure 5. Storage comparison details are in Appendix C.

**Ablation Study.** We conducted ablation experiments on the parameter decoupling steps to evaluate their effectiveness. In Table 5, we begin with the original parameter decoupling strategy and systematically remove components such as Directional Congruence, Dominant Significance, and the scaling factor $\lambda$. We then report the performance of MMER in both multi-modality expansion and retention scenarios. Removing Directional Congruence means that parameters are selected solely based on Dominant Significance, i.e., $m_i = 1\{ |\tau_i| \geq 50\% \cdot \lambda_i |\tau_*| \}$. Removing Dominant Significance entails retaining parameters solely based on the consistency of their signs, i.e., $m_i = 1\{sign(\tau_i) = sign(\tau_*)\}$. Table 5 shows that the steps in parameter decoupling are crucial for optimizing performance. Specifically, Directional Congruence proves to be the most critical. Without it, the decoupled parameters are almost meaningless and lose all the original modality information. Next in importance is Dominant Significance. Without filtering out crucial parameters, irrelevant ones persist and significantly interfere with the original parameters. Finally, the scaling factor $\lambda$ also plays a role, causing performance degradations of 4.93% in multi-modality expansion experiments and 4.92% in retention experiments.

Table 5: Ablation study on all steps of parameter decoupling. In retention experiments, we excluded three point and audio classification tasks.

| Method | 3 Expansion Tasks Avg ACC. | 11 Retention Tasks Avg Score (%) / Avg ACC. (%) |
|---|---|---|
| **MMER** | **56.82** | **24.17** $_{(99.8)}$ / **50.84** $_{(99.4)}$ |
| − Directional Congruence | 7.20 | 10.05 $_{(41.6)}$ / 8.34 $_{(16.7)}$ |
| − Dominant Significance | 33.87 | 14.71 $_{(60.5)}$ / 28.93 $_{(57.1)}$ |
| − Scaling Factor $\lambda$ | 54.02 | 23.14 $_{(95.6)}$ / 47.78 $_{(93.9)}$ |

# 7 CONCLUSION

In this paper, we propose a novel training-free approach MMER that reuses and composes existing MLLMs to resolve the dilemma faced by MLLMs during multimodal expansion: costly retraining or suboptimal performance. MMER retains multimodal encoders and merges LLM parameters, constructing binary masks to decouple modality-specific parameters. This innovative mechanism enables independent handling of modality-specific inputs by their respective parameters, thereby significantly reducing parameter conflicts to enhance multimodal performance. Additionally, MMER can reconstruct the original MLLMs by integrating decoupled parameters with the base LLM, effectively retaining the original performance. Finally, by incorporating fine-tuned MLLMs for new tasks into MMER approach, it isolates the parameters for new tasks from the original ones, thus mitigating catastrophic forgetting. Extensive experiments have validated the effectiveness and robustness of MMER. We hope this work inspires further exploration of training-free multimodal expansion for LLMs.

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

## A   NOVELTY AND CONTRIBUTIONS

Our research aims to achieve training-free multi-modality expansion and retention for LLMs through parameter merging and decoupling. We conduct a comparative analysis with existing relevant methods to demonstrate the innovation of our MMER approach.

**Comparison with NaiveMC and DAMC frameworks.** Our MMER approach is based on the NaiveMC framework (Chen et al., 2024) and employs a parameter dynamic decoupling strategy similar to that of the DAMC framework (Chen et al., 2024) to mitigate parameter conflicts in the merged MLLM. However, there are several key differences:

1. Compared to the NaiveMC framework, our MMER approach effectively enhances the multimodal performance of the merged MLLM.

2. Compared to the DAMC framework, our MMER approach employs a training-free parameter decoupling strategy instead of separating parameters during the initialization training of the MLLMs and achieves similar results. Additionally, MMER is additional compatible with full-parameter fine-tuned MLLMs, whereas DAMC is restricted to parameter-efficient fine-tuned MLLMs.

3. Compared to the NaiveMC and DAMC frameworks, our MMER approach retains the performance of the original MLLMs while also providing additional capabilities to mitigate catastrophic forgetting.

Our MMER approach integrates the strengths of the NaiveMC and DAMC frameworks, while additionally providing original performance retention capabilities.

**Comparison with training-free model merging methods.** Training-free model merging methods, such as TA (Ilharco et al., 2023), TIES (Yadav et al., 2023), and DARE (Yu et al., 2024), are primarily designed for merging models with identical architectures. Consequently, they must be combined with the NaiveMC framework to achieve multi-modality expansion for LLMs. These methods alleviate parameter conflicts in merged MLLMs to some extent, leading to performance enhancement. However, their overall effectiveness, both in terms of multimodal performance and retention of original performance, falls significantly short compared to our MMER approach.

**Comparison with alignment and fine-tuning methods.** Compared to methods (Chen et al., 2023a; Lyu et al., 2023; Han et al., 2024) that achieve multimodal expansion for LLMs by adding multiple new modality encoders or employing a unified multimodal encoder followed by alignment and fine-tuning, the advantages of our MMER approach are clear. MMER can effectively reuse a large number of MLLMs from the open-source community and merge them enabling multimodal expansion without the need for extensive resources and time spent on training models and constructing high-quality modality instruction data.

**Comparison with TALL-masks.** TALL-masks (Wang et al., 2024) is an information localization algorithm that, similar to our approach, compresses original parameters and subsequently approximates their restoration. However, there are several key differences:

1. From an algorithmic perspective, TALL-masks overlooks the Consistency of original and merged parameter signs. In contrast, we have addressed this aspect and demonstrated its effectiveness in our ablation experiments (See Table 5).

2. In terms of application scenarios, our MMER applies parameter merging and decoupling to the multimodal expansion for LLMs, enhancing their multimodal capabilities. Additionally, we utilize MMER to mitigate catastrophic forgetting. These aspects are not considered by TALL-masks.

3. Regarding the models utilized, the models used in our MMER approach are the 7B MLLMs across various modalities, while TALL-masks is applied to relatively smaller models within the same modality, such as T5 (Raffel et al., 2020) and ViT (Dosovitskiy et al., 2021).

## B    IMPLEMENTATION AND EXPERIMENTAL DETAILS

All our experiments are conducted on an NVIDIA 8×A800-SXM4-80GB machine.

### B.1    PERFORMANCE RETENTION

Considering the varying modalities of each original MLLM and the different evaluation metrics for distinct tasks, we provide performance retention in our results to validate the method's capacity to retain original performance. The definition is as follows:

$$\text{Performance Retention} = \frac{1}{T} \sum_{t=1}^{T} \frac{\underset{x\sim\mu_t}{\text{metric}} \left[ f_{\text{method}}(x) \right]}{\underset{x\sim\mu_t}{\text{metric}} \left[ f_{\text{original}}(x) \right]} \tag{8}$$

The "metric" refers to various evaluation metrics, such as accuracy and captioning scores(e.g., BLEU, ROUGE).

### B.2    IMPLEMENTATION DETAILS OF ORIGINAL FINE-TUNED MLLMS

We utilize the same training data and components of each MLLM across the four modalities following NaiveMC (Chen et al., 2024). More details are presented in Table 6.

Table 6: Training data and components of MLLMs for different modalities.

| Modality | Modality Encoder | Connector | Alignment Data | Fine-tuneing Data | Referenced Work |
|---|---|---|---|---|---|
| Image | CLIP-ViT-L-336px (Dosovitskiy et al., 2021) | MLP | LCS 558K (Xu et al., 2024) | LLaVA-mixed 665K (Xu et al., 2024) | LLaVA-1.5 (Liu et al., 2024) |
| Audio | BEATs-Iter3+ (Chen et al., 2023b) | Q-Former | WaveCaps 400K (Mei et al., 2024) | OpenAQA filtered 350K (Gong et al., 2024) | X-InstructBLIP (Panagopoulou et al., 2023) |
| Video | LanguageBind (Zhu et al., 2024a) | MLP | LCS 558K, Valley 702K (Luo et al., 2023) | Video-ChatGPT 100K (Maaz et al., 2024), LLaVA-mixed sampled 140K | Video-LLaVA (Lin et al., 2023) |
| Point Cloud | Point Encoder (Xu et al., 2024) | MLP | PointLLM brief description 660K (Xu et al., 2024) | Point complex instruction 70K (Xu et al., 2024) | PointLLM (Xu et al., 2024) |

We adopt similar hyperparameters following previous works (Chen et al., 2024; Liu et al., 2024; Panagopoulou et al., 2023; Lin et al., 2023; Xu et al., 2024). During the alignment stage, only the parameters in the connectors are trainable. In the fine-tuning stage, we tune all connector parameters and base LLM parameters. For training efficiency, we utilize DeepSpeed Zero Optimization Stage 3. Detailed data are presented in the Table 7.

### B.3    BASELINE DETAILS

In this section, we provide a detailed overview of the six baselines included in our experiments:

- **Original MLLMs** means that each MLLM is evaluated on its corresponding modality benchmarks to demonstrate its original performance, but they cannot perform cross-modal tasks simultaneously.

- **NaiveMC framework** (Chen et al., 2024) combines modality-specific encoders from multiple MLLMs into the merged LLM, which is obtained by averaging the parameters of multiple LLMs from these MLLMs. The averaging merging strategy can be replaced by other model merging methods.

- **TA** (Ilharco et al., 2023) initially defines the concept of *task vector* and employs arithmetic operations for model merging, model forgetting, and support multi-task learning, etc. The final model is formed by scaling and adding task vectors to the initial model, represented mathematically as $\theta_m = \theta_{\text{init}} + \lambda \cdot \sum_{t=1}^{n} \tau_t$.

Table 7: Hyperparameters of different MLLMs.

| Stage | Hyperparameter | Image | Audio | Video | Point Cloud |
|---|---|---|---|---|---|
| Alignment-State | Batch size | 256 | 256 | 256 | 128 |
| | LR | 1e-3 | 1e-3 | 1e-3 | 2e-3 |
| | LR Schedule | | cosine decay | | |
| | Warmup Ratio | | 0.03 | | |
| | Epoch | 1 | 1 | 1 | 3 |
| Fine-tuning-Stage | Batch size | 128 | 64 | 128 | 64 |
| | LR | 2e-5 | 1e-5 | 2e-5 | 2e-5 |
| | LR Schedule | | cosine decay | | |
| | Warmup Ratio | | 0.03 | | |
| | Epoch | 1 | 3 | 1 | 3 |

- **TIES** (Yadav et al., 2023) improves upon TA (Ilharco et al., 2023) by further mitigating parameter interference. It first prunes redundant parameters to retain the most important ones. When encountering conflicts in parameter signs during merging, it selects and merges parameters with the majority sign while ignoring those with minority signs.

- **DARE** (Yu et al., 2024) proposes a preprocessing step to address parameters conflict. This method randomly discards the majority of the delta parameters while scaling the remaining ones by $\theta' = \theta \cdot (1/(1-p))$ where $p$ is the proportion of dropped delta parameters.

- **Model Tailor** (Zhu et al., 2024c) identifies the key parameters fine-tuned on the new tasks within the MLLM and integrates them into the original MLLM, thereby retaining the performance on previous tasks while adapting to new tasks.

## C    STORAGE COST CALCULATION

This section details the calculation of storage costs for MMER approach and the relevant methods mentioned above. Let $N$, $P$, $P'$, and $P^*$ represent the number of original MLLMs, the total parameters of the LLMs, the number of the modality-specific component parameters, and the number of additional trainable parameters of parameter-efficient fine-tuning methods, respectively. Assuming each float parameter occupies 32 bits, the storage cost for these methods across $N$ original MLLMs is calculated as follows:

- Original fine-tuned models: $32N(P+P')$. $32(P+P')$ represents the number of parameters contained in a single MLLM.

- NaiveMC framework: $32P + 32NP'$. Stores a merged LLM and $N$ modality-specific components.

- DAMC framework: $32P + 32NP' + 2N(32P^*)$. Stores a merged LLM and $N$ modality-specific components. $2N(32P^*)$ represents the need to store an additional $2N$ trainable parameters of parameter-efficient fine-tuning methods for parameter separation.

- NaiveMC wit TA / TIES / DARE: $32P + 32NP'$. Same as the NaiveMC framework.

- MMER: $64P + 32NP' + NP$. $64P$ is for storing the parameters of a base LLM and a merged task vector, while $32NP'$ indicates $N$ modality-specific components. Additionally, $NP$ denotes the storage for $N$ binary masks.

## D    MORE DETAILS ABOUT EXPERIMENTS

### D.1    MORE RESULTS

We supplement the results of mainstream MLLMs, such as ImageBind-LLM and OneLLM, on multimodal benchmarks in Table 8 and observe that MMER achieves superior performance compared to them.

### D.2    DETAILED RESULTS

In this section, we present detailed results from the multi-modality retention and mitigating catastrophic forgetting experiments. The results of various baselines for seven image tasks are shown in Table 9, two point cloud tasks in Table 10, three audio tasks and two video tasks in Table 11, three multimodal tasks in Table 12, and the last two new tasks in Table 13.

Table 8: The results of mainstream MLLMs on multimodal benchmarks

| Task (→) | ModelNet40 | MUSCI-AVQA | | | MCUB | | | | | Avg |
|---|---|---|---|---|---|---|---|---|---|---|
| Method (↓) | PI-T | IA-T | VI-T | VA-T | AVI-T | AVP-T | AIP-T | VIP-T | AVIP-T | |
| OneLLM-7B (Han et al., 2024) | - | 44.64 | 45.48 | 47.60 | - | - | - | - | - | - |
| ImageBind-LLM(Han et al., 2023) | 39.86 | 36.54 | 38.76 | 39.72 | 35.20 | 31.40 | 33.40 | 31.80 | 32.93 | 35.51 |
| X-InstructBLIP(Panagopoulou et al., 2023) | 57.93 | 40.71 | 41.23 | 48.34 | 41.40 | 25.20 | 21.20 | 29.40 | 27.94 | 37.04 |
| **MMER (ours)** | **62.15** | **47.25** | **51.27** | **51.77** | **56.48** | **59.31** | **65.59** | **56.00** | **61.63** | **56.82** |

Table 9: Results for each method on seven image tasks. All tasks are Question-Answering tasks.

| Task (→) | 7 Image Tasks | | | | | | |
|---|---|---|---|---|---|---|---|
| | VQAv2 | GQA | TextVQA | VizWiz | ScienceQA | POPE | OK-VQA |
| Method (↓) | Acc. | Acc. | Acc. | Acc. | Acc. | Acc. | Acc. |
| Original MLLMs | 78.11 | 61.52 | 55.89 | 51.51 | 71.12 | 86.17 | 31.33 |
| **MMER (ours)** | 77.95 | 61.85 | 55.74 | 52.26 | 71.16 | 86.58 | 31.27 |
| *–Multi-Modality Retention* | | | | | | | |
| NaiveMC [ACL2024] (Chen et al., 2024) | 59.73 | 45.83 | 42.29 | 47.87 | 68.52 | 79.41 | 24.28 |
| TA [ICLR23] (Ilharco et al., 2023) | 62.71 | 48.86 | 45.20 | 49.47 | 70.04 | 82.38 | 25.56 |
| TIES [NeurIPS23] (Yadav et al., 2023) | 61.78 | 48.23 | 44.60 | 48.67 | 69.05 | 81.21 | 25.13 |
| NaiveMC (w/ DARE[ICML2024] (Yu et al., 2024)) | 60.91 | 46.62 | 42.88 | 49.04 | 70.09 | 81.08 | 24.62 |
| TA (w/ DARE) | 63.65 | 49.25 | 45.74 | 49.82 | 70.87 | 83.12 | 25.82 |
| TIES (w/ DARE) | 62.54 | 48.73 | 45.38 | 49.15 | 69.78 | 82.17 | 25.39 |
| *–Mitigating Catastrophic Forgetting* | | | | | | | |
| Fine-tune on Flickr30k | 72.27 | 54.19 | 46.10 | 52.88 | 70.22 | 76.78 | 28.31 |
| **MMER-Clotho-AQA** | 77.87 | 61.59 | 55.51 | 51.88 | 71.16 | 86.24 | 31.14 |
| **MMER-Flickr30k** | 77.75 | 61.43 | 55.41 | 52.72 | 71.75 | 85.72 | 31.07 |
| **MMER-Clotho-AQA+Flickr30k** | 77.32 | 61.33 | 55.23 | 52.33 | 71.02 | 85.43 | 30.94 |

# E  QUALITATIVE RESULTS

We provide qualitative results in Figure 6 and 7. These results demonstrate the capability of the merged MLLM constructed by our MMER approach to understand and reason with multimodal inputs.

# F  PROMPT FOR EVALUATION

We present the evaluation prompts for each benchmark in Table 14. To denote the inputs for various modalities, we use "<image>", "<audio>", "<video>", and "<point>" to represent image, audio, video, and point cloud modalities, respectively.

Table 10: Results for each method on two point cloud tasks. Among them, ModelNet40 is a classification task, while Objavers is a captioning task.

| Task (→) | 2 Point Tasks | | | | | |
|---|---|---|---|---|---|---|
| | ModelNet40 | Objavers | | | | |
| Method (↓) | Acc. | BLEU-1 | ROUGE-L | METEOR | Sentence-BERT | SimCSE |
| Original MLLMs | 21.27 | 4.73 | 8.51 | 12.02 | 44.18 | 46.31 |
| **MMER (ours)** | 22.49 | 5.06 | 8.53 | 11.90 | 43.72 | 46.51 |
| *–Multi-Modality Retention* | | | | | | |
| NaiveMC [ACL2024] (Chen et al., 2024) | 20.49 | 4.43 | 8.24 | 11.37 | 43.22 | 45.97 |
| TA [ICLR23] (Ilharco et al., 2023) | 21.02 | 4.69 | 8.46 | 11.73 | 43.55 | 46.38 |
| TIES [NeurIPS23] (Yadav et al., 2023) | 20.83 | 4.55 | 8.39 | 11.60 | 43.29 | 46.27 |
| NaiveMC (w/ DARE[ICML2024] (Yu et al., 2024)) | 20.77 | 4.41 | 8.38 | 11.59 | 43.47 | 46.28 |
| TA (w/ DARE) | 21.25 | 4.81 | 8.49 | 11.82 | 43.67 | 46.42 |
| TIES (w/ DARE) | 20.98 | 4.62 | 8.31 | 11.47 | 43.14 | 46.28 |
| *–Mitigating Catastrophic Forgetting* | | | | | | |
| **MMER-Clotho-AQA** | 21.87 | 4.92 | 8.46 | 11.52 | 43.55 | 46.28 |
| **MMER-Flickr30k** | 22.03 | 5.08 | 8.55 | 11.63 | 43.61 | 46.36 |
| **MMER-Clotho-AQA+Flickr30k** | 21.56 | 4.98 | 8.39 | 11.38 | 43.34 | 46.02 |

Table 11: Results for each method on three audio tasks and two video tasks. Among them, TUT, VocalSound, MSVD, and MSRVTT are the classification tasks, while Clotho is a captioning task.

| Task (→) | 3 Audio Tasks | | | | | 2 Video Tasks | |
|---|---|---|---|---|---|---|---|
| | TUT | VocalSound | Clotho | | | MSVD | MSRVTT |
| Method (↓) | Acc. | Acc. | CIDEr | SPICE | SPIDEr | Acc. | Acc. |
| Original MLLMs | 22.23 | 27.19 | 38.63 | 11.98 | 25.29 | 48.40 | 31.18 |
| **MMER (ours)** | 34.14 | 42.88 | 38.49 | 11.93 | 25.18 | 48.12 | 30.43 |
| *–Multi-Modality Retention* | | | | | | | |
| NaiveMC [ACL2024] (Chen et al., 2024) | 29.50 | 31.80 | 37.56 | 11.61 | 24.61 | 44.53 | 29.31 |
| TA [ICLR23] (Ilharco et al., 2023) | 30.64 | 33.12 | 37.69 | 11.67 | 24.69 | 45.61 | 29.54 |
| TIES [NeurIPS23] (Yadav et al., 2023) | 30.87 | 33.42 | 37.89 | 11.72 | 24.78 | 45.88 | 29.74 |
| NaiveMC (w/ DARE[ICML2024] (Yu et al., 2024)) | 30.50 | 32.75 | 37.75 | 11.66 | 24.74 | 45.69 | 29.58 |
| TA (w/ DARE) | 30.98 | 33.90 | 37.87 | 11.69 | 24.89 | 45.51 | 29.54 |
| TIES (w/ DARE) | 31.59 | 34.45 | 37.96 | 11.87 | 24.92 | 46.07 | 29.93 |
| *–Mitigating Catastrophic Forgetting* | | | | | | | |
| Fine-tune on Clotho-AQA | 6.98 | 17.65 | 30.02 | 9.40 | 20.04 | - | - |
| **MMER-Clotho-AQA** | 34.01 | 42.45 | 38.37 | 11.89 | 25.11 | 48.04 | 30.29 |
| **MMER-Flickr30k** | 33.41 | 41.94 | 38.10 | 11.81 | 24.98 | 47.74 | 30.05 |
| **MMER-Clotho-AQA+Flickr30k** | 33.54 | 41.83 | 37.97 | 11.76 | 24.92 | 47.38 | 29.67 |

Table 12: Results of the mitigating catastrophic forgetting experiments for three MMER variants on multimodal tasks with different combinations of video (V), image (I), audio (A), point cloud (P), and text (T) inputs.

| Task (→) | ModelNet40 | MUSCI-AVQA | | | MCUB | | | | |
|---|---|---|---|---|---|---|---|---|---|
| Method (↓) | PI-T | IA-T | VI-T | VA-T | AVI-T | AVP-T | AIP-T | VIP-T | AVIP-T |
| **MMER-Clotho-AQA** | 61.98 | 47.01 | 51.22 | 51.43 | 56.08 | 59.11 | 65.08 | 55.80 | 61.08 |
| **MMER-Flickr30k** | 61.84 | 46.92 | 51.05 | 51.56 | 56.28 | 58.90 | 65.08 | 55.40 | 60.93 |
| **MMER-Clotho-AQA+Flickr30k** | 61.33 | 46.48 | 50.61 | 51.17 | 55.68 | 57.93 | 64.17 | 55.20 | 60.42 |

Table 13: Results of the mitigating catastrophic forgetting experiments for each method on two **new tasks**. Among them, Clotho-AQA is a Question-Answering task, while Flickr30k is a captioning task.

| Task (→) | Clotho-AQA | Flickr30k | | | | | | | |
|---|---|---|---|---|---|---|---|---|---|
| Method (↓) | Acc. | CIDEr | METEOR | BLEU-1 | BLEU-2 | BLEU-3 | BLEU-4 | ROUGE | |
| Original MLLMs | 49.40 | 80.27 | 25.62 | 73.29 | 55.81 | 41.11 | 29.66 | 53.11 | |
| Fine-tune on Clotho-AQA | 57.80 | - | - | - | - | - | - | - | |
| Fine-tune on Flickr30k | - | 94.25 | 27.74 | 78.27 | 62.24 | 47.99 | 36.50 | 57.04 | |
| **MMER (ours)** | 49.28 | 79.56 | 25.56 | 73.11 | 55.62 | 40.83 | 29.49 | 52.82 | |
| **MMER-Clotho-AQA** | 57.71 | 79.72 | 25.51 | 73.04 | 55.48 | 40.72 | 29.33 | 52.77 | |
| **MMER-Flickr30k** | 48.94 | 92.74 | 27.58 | 77.93 | 61.69 | 47.36 | 36.02 | 56.27 | |
| **MMER-Clotho-AQA+Flickr30k** | 57.52 | 92.09 | 27.29 | 77.22 | 61.11 | 46.86 | 35.62 | 55.72 | |

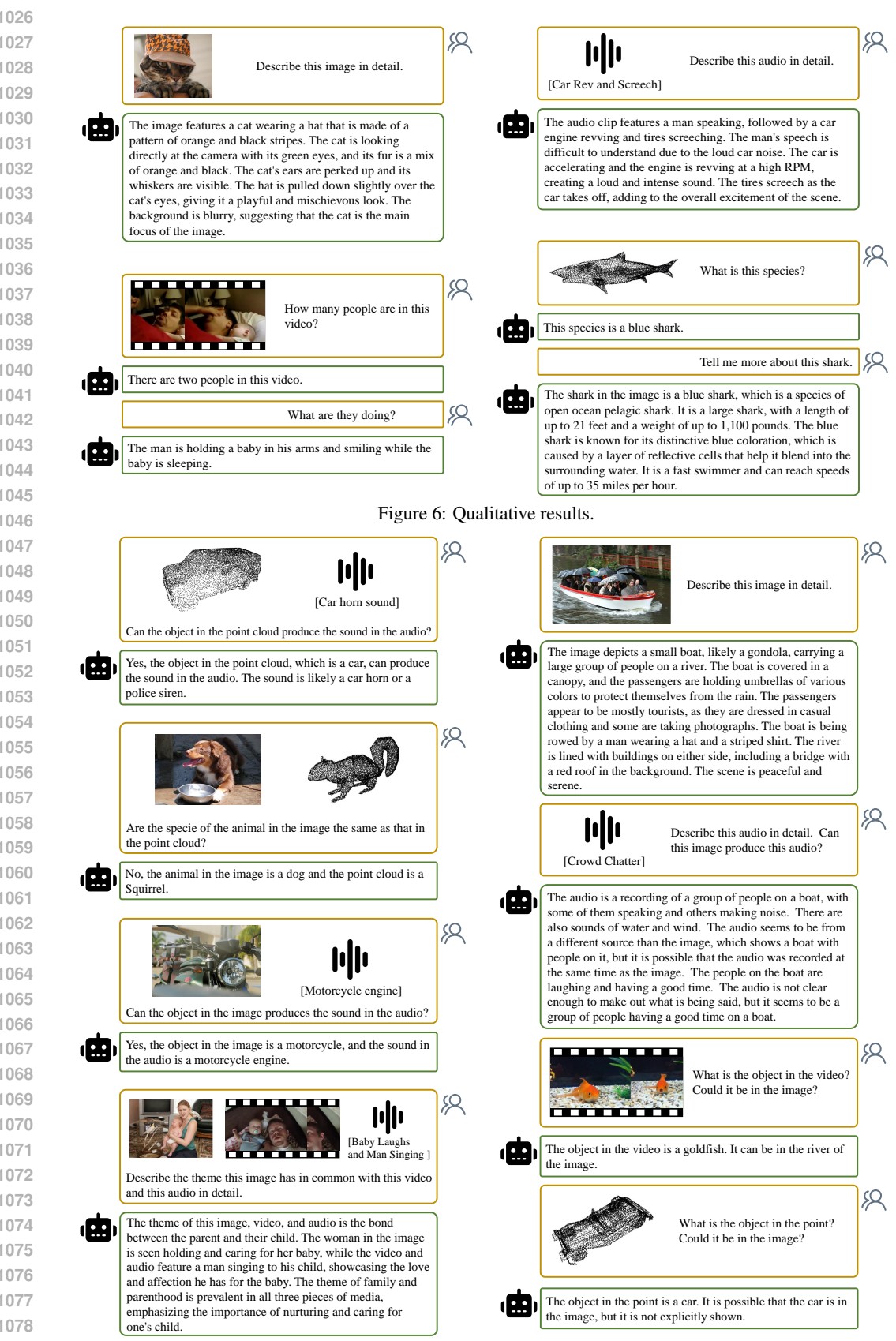

Figure 6: Qualitative results.

Figure 7: Qualitative results.

Table 14: Prompt Template for different evaluation benchmarks.

| Benchmark | Modality | Prompt Template |
|---|---|---|
| MCUB | AVI-T | Based on four input entities:\nimage <image>\naudio <audio>\nvideo <video>\n {Question} {Options} Answer with the option's letter from the given choices directly. |
| | AVP-T | Based on four input entities:\naudio <audio>\nvideo <video>\npoint <point>\n {Question} {Options} Answer with the option's letter from the given choices directly. |
| | VIP-T | Based on four input entities:\nimage <image>\nvideo <video>\npoint <point>\n {Question} {Options} Answer with the option's letter from the given choices directly. |
| | AIP-T | Based on three input entities:\nimage <image>\naudio <audio>\npoint <point>\n {Question} {Options} Answer with the option's letter from the given choices directly. |
| | AVIP-T | Based on four input entities:\nimage <image>\naudio <audio>\nvideo <video>\npoint <point>\n {Question} {Options} Answer with the option's letter from the given choices directly. |
| MUSIC-AVQA | VI-T | Based on the video <video> and image <image>\n{Question} \nAnswer the question using a single word. |
| | VA-T | Based on the video <video> and audio <audio>\n{Question} \nAnswer the question using a single word. |
| | IA-T | Based on the image <image> and audio <audio>\n{Question} \nAnswer the question using a single word. |
| ModelNet40 | PI-T | Based on rendered image <image> and point cloud <point>\nWhat is this? Select from these objects: {Options} Answer the question using a single word. |
| | I-T | <point>\nWhat is this? Select from these objects: {Options} Answer the question using a single word. |
| Objaverse | I-T | <point>\nOffer a clear and concise description of this point cloud object. |
| VocalSound & TUT | A-T | <audio>\nWhich of the following categories does this audio belong to? {Options} Answer the question using a single word. |
| Clotho | A-T | <audio>\nDescribe this audio in detail. |
| Clotho-AQA | A-T | <audio>\n{Question}\nAnswer the question using a single word or phrase. |
| MSRVTT & MSVD | V-T | <video>\n{Question}\nAnswer the question using a single word or phrase. |
| VQAv2 & GQA & POPE & OK-VQA | I-T | <image>\n{Question}\nAnswer the question using a single word or phrase. |
| Textvqa | I-T | <image>\n{Question}\nReference OCR token: {Options}\nAnswer the question using a single word or phrase. |
| VizWiz | I-T | <image>\n{Question}\nWhen the provided information is insufficient, respond with 'Unanswerable'.\nAnswer the question using a single word or phrase. |
| ScienceQA | I-T | <image>\n{Context}\n{Question}\nChoose the most likely ratio. {Options} |
| Flickr30k | I-T | <image>\nDescribe this image using one or more simple sentences. |

