# OpenReview forum: "Multi-modality Expansion and Retention for LLMs through Parameter Merging and Decoupling"
_ICLR.cc/2025/Conference — Submitted to ICLR 2025_

### Official Review · Reviewer_egSN · 2024-11-03

**Soundness:** 3
**Presentation:** 3
**Contribution:** 2
**Rating:** 5
**Confidence:** 3

**Summary:**

This paper introduces a novel training-free approach called MMER (Multi-modality Expansion and Retention). The approach aims to expand the capabilities of Large Language Models (LLMs) to handle multimodal data while retaining their original performance on specific modalities.
MMER merges LLM parameters from multiple MLLMs while constructing binary masks to decouple modality-specific parameters. This allows for independent processing of modality-specific inputs and reduces interference.

**Strengths:**

The paper introduces a new parameter decoupling strategy that effectively separates modality-specific parameters, reducing interference and enhancing performance.

The authors have clearly articulated the problem, the proposed solution, and the advantages of MMER over existing methods.

The paper is clear and well-organized, making it easy for readers to follow the authors' line of reasoning.

**Weaknesses:**

In general, cross-modal tasks exhibit diversity, and the paper reports performance on only one cross-modal task for each modality combination, such as MUSIC-AVQA and MCUB. This makes the validation of the proposed method's effectiveness insufficiently comprehensive.

Lack of the performance of the unimodal model on the tri-modality task for a fairer comparison.

The author decouples modalities using addition and subtraction; however, models are typically nonlinear, which confuses me. The paper also lacks sufficient proof for this approach. Additionally, the justification for using Manhattan distance when calculating the mask is not strong enough.

**Questions:**

Can the author clarify how many parameters are decoupled for each modality?

I hope the author can provide a more detailed explanation and proof of the proposed method. I will consider changing my perspective if the author's response is valuable.

---

> ### Author Response · Authors · 2024-11-20
> **Response to Reviewer egSN (1/2)**
>
> Thank you for your valuable comments. We will address your concerns point by point.
>
> > **Comment1**: In general, cross-modal tasks exhibit diversity, and the paper reports performance on only one cross-modal task for each modality combination, such as MUSIC-AVQA and MCUB.
>
> **Reply**:
>
> - For the **PI-T** modality combination, we have additionally provided experimental results on the Objaverse classification task.
>
> - For the **IA-T**, **VI-T**, and **VA-T** modality combinations, we have supplemented results on the AVQA dataset.
>
> - For the **AVI-T** modality combination, we have added results on both the AVQA and MUSIC-AVQA datasets.
>
> - As for other modality combinations, further experiments could not be conducted due to the constraints of currently available datasets.
>
>   | Task (→)          | Objaverse | MUSCI-AVQA | AVQA  | AVQA  | AVQA  | AVQA  |
>   | ----------------- | :-------: | :--------: | :---: | :---: | :---: | :---: |
>   | **Model (↓)**     |   PI-T    |   AVI-T    | IA-T  | VI-T  | VA-T  | AVI-T |
>   | NaiveMC           |   57.76   |   49.64    | 72.63 | 74.12 | 74.27 | 75.06 |
>   | TA                |   58.97   |   49.93    | 74.75 | 76.34 | 76.58 | 77.25 |
>   | TIES              |   59.27   |   51.19    | 75.84 | 77.69 | 77.88 | 78.24 |
>   | NaiveMC (w/ DARE) |   58.02   |   49.89    | 73.66 | 75.17 | 75.22 | 76.16 |
>   | TA (w/ DARE)      |   59.55   |   50.27    | 75.52 | 77.24 | 77.47 | 78.03 |
>   | TIES (w/ DARE)    |   60.04   |   51.36    | 76.24 | 77.95 | 78.14 | 78.51 |
>   | **MMER (ours)**  |   59.87   |   53.54    | 78.13 | 79.74 | 79.98 | 80.72 |
>
>
>
> > **Comment2**: Lack of the performance of the unimodal model on the tri-modality task for a fairer comparison.
>
> **Reply**: We have added the results of four original unimodal models on the tri-modality task. By referring to **Table 1**, we can observe that MMER consistently outperforms unimodal models. This advantage is not only attributed to the incorporation of additional modality information but, more importantly, to its ability to approximately decouple modality information, thereby reducing conflicts between modality parameters. In contrast, other model merging methods fail to consistently surpass the performance of unimodal models.
>
> | Task/Model           | ModelNet40 | MUSCI-AVQA |
> | -------------------- | :--------: | :--------: |
> | original Point MLLM  |   21.27    |     -      |
> | original Vision MLLM |   51.94    |   44.06    |
> | original Audio MLLM  |     -      |   30.63    |
> | original Video MLLM  |     -      |   47.72    |
>
>
>
> > **Comment3**: The author decouples modalities using addition and subtraction; however, models are typically nonlinear, which confuses me. The paper also lacks sufficient proof for this approach.
>
> **Reply**: In the initial study, TA [2] introduced the concept of *task arithmetic*, proposing that “we can edit a variety of models with task arithmetic—performing simple arithmetic operations on task vectors (i.e., $θ_{ft}−θ_{pre}$, representing a direction in the pre-trained model's weight space such that moving along this direction improves performance on the corresponding task).” They observed that arithmetic operations between task vectors often result in analogous functional behaviors. For instance, summing the task vectors of a model pre-trained and fine-tuned on two separate tasks yields a new multi-task model with enhanced performance on both tasks. Since then, numerous studies [5,6] based on task arithmetic have emerged, and their successes serve as indirect evidence of the effectiveness of this simple arithmetic operation.
>
> Additionally, TA [2] explored the cosine similarity between task vectors across different tasks. They observed that vectors from distinct tasks are typically close to orthogonal, suggesting that this orthogonality facilitates the combination of task vectors via addition with minimal interference. Moreover, they found higher cosine similarities when tasks are semantically similar, which partly explains why simple arithmetic operations can effectively merge and disentangle task-specific capabilities within models.
>
> Subsequent research [1] delved deeper into the principles underlying task arithmetic and provided new insights. They demonstrated that task arithmetic is possible thanks to the fact that models inherently operate in a linear regime, where their behavior is dictated by the finite-width neural tangent kernel (NTK). They further proved that the sole requirement for task arithmetic is actually weight disentanglement, where distinct directions (i.e., task vectors) in weight space correspond to changes of the network in disjoint regions of the input space. This allows a model to perform task arithmetic by independently manipulating these weight directions.
>
>
>
> **Reference**
>
> [1] Ortiz-Jimenez et al. Task Arithmetic in the Tangent Space: Improved Editing of Pre-Trained Models. NeurIPS, 2023.
>
> [2] Ilharco et al. Editing models with task arithmetic. ICLR, 2023.

---

> ### Author Response · Authors · 2024-11-20
> **Response to Reviewer egSN (2/2)**
>
> > **Comment4**: The justification for using Manhattan distance when calculating the mask is not strong enough.
>
> **Reply**: First, for the sake of saving computational resources and improving efficiency, we did not adopt methods like Fisher [3] or Regmean [4], which require additional gradient-based computations to obtain the Information Matrix, as these methods demand substantial computational resources or data.
>
> Inspired by TIES [6] and DARE [5], which propose that “Supervised fine-tuned language models tend to acquire excessively redundant delta parameters”, we aim to decouple the most critical parameter of each modality from the merged task vector, so that the decoupled parameters are as close as possible to the original modality task vector.
>
> Considering the need to save storage and reduce computational resources, and based on DARE's [5] findings that key parameters are highly sparse, we decided to use a binary mask matrix to directly mask out irrelevant parameters in the merged task vector and only retain the key information related to the modality.
>
> We chose to use the Manhattan distance to optimize the mask mainly due to its mathematical properties and its promotion of sparsity in high-dimensional parameter spaces. Here are some detailed reasons and explanations:
>
> First, **Manhattan distance naturally facilitates parameter sparsification** because it tends to drive parameters to zero, which aligns perfectly with the binary mask matrix we aim to construct. The goal of the mask is to select key parameters relevant to a specific modality from the merged task vector and mask out irrelevant ones, meaning that most elements in the mask should be zero, with only a few elements set to 1. By minimizing the Manhattan distance, we can easily achieve this goal because the gradient of parameter updates with respect to Manhattan distance is constant. This makes it more likely to penalize smaller non-zero parameters and drive them to zero, thus encouraging the sparsity of the mask. Moreover, these smaller non-zero parameters are often redundant [6], which are the ones we wish to mask out.
>
> In contrast, Euclidean distance squares each difference, which results in larger magnitude parameter changes being penalized more significantly. **Therefore, it tends to shrink larger magnitude parameters, rather than smaller non-zero ones, which makes the mask tend towards smoothness rather than sparsity.** However, larger magnitude parameters are typically key and modality-specific [6], and this smoothing characteristic does not align with our need to construct the mask matrix. This smoothing effect could lead to the loss of critical parameter information. Hence, using Manhattan distance more effectively promotes the sparsity of the mask, extracting a modality-specific parameter subset.
>
> Furthermore, **Manhattan distance directly measures the element-wise difference between the merged task vector and the original modality task vectors**. This element-wise comparison can precisely capture which parameters have undergone significant changes during fine-tuning and which parameters are irrelevant noise. In contrast, **Euclidean distance emphasizes the “total distance” in the overall parameter space and fails to fully reflect the contribution of individual parameters**, making it less direct and effective than Manhattan distance in constructing the mask matrix.
>
> We have also conducted both the multimodal expansion and retention experiments that validate the effectiveness of Manhattan distance over Euclidean distance:
>
> |                               | 3 Expansion Tasks |  11 Retention Tasks  |
> | ----------------------------- | :---------------: | :------------------: |
> |                               |     Avg ACC.      | Avg Score / Avg ACC. |
> | **MMER (Manhattan distance)** |       56.82       |    24.17 / 50.84     |
> | **MMER (Euclidean distance)** |       56.05       |    23.89 / 50.41     |
>
>
>
> > **Question1**: Can the author clarify how many parameters are decoupled for each modality?
>
> **Reply**: In **Figure 4a** of our paper, we illustrate the percentage of parameters decoupled for each modality. Specifically, the audio mask, point mask, video mask, and image mask decouple 2.2%, 43.8%, 15.7%, and 34.1% of the parameters, respectively. Additionally, regarding why the audio mask decouples  only 2.2% of the parameters, we have provided further clarification and explanation in our response to reviewer VHBG. We recommend referring to that section for more detailed information on these results.
>
>
>
> **Reference**
>
> [3] Matena et al. Merging Models with Fisher-Weighted Averaging. NeurIPS, 2022.
>
> [4] Jin et al. Dataless knowledge fusion by merging weights of language models. ICLR, 2023.
>
> [5] Yu et al. Language Models are Super Mario: Absorbing Abilities from Homologous Models as a Free Lunch. ICML, 2024.
>
> [6] Yadav et al. Ties-merging: Resolving interference when merging models. NeurIPS, 2023.

---

### Official Review · Reviewer_Q9ea · 2024-11-03

**Soundness:** 3
**Presentation:** 3
**Contribution:** 3
**Rating:** 6
**Confidence:** 4

**Summary:**

- The paper proposes MMER (Multi-modality Expansion and Retention), a novel training-free approach that reuses and composes existing MLLMs to facilitate effective multimodal expansion while retaining the original performance of each MLL
- Extensive and rigorous experiments on various multimodal benchmarks have been conducted to validate the effectiveness and robustness of MMER.

**Strengths:**

- The paper proposes a training-free MMER approach that enables seamless multimodal expansion for LLMs through multimodal parameter merging and decoupling.
- The paper also leverages MMER for mitigating catastrophic forgetting in MLLMs, demonstrating its potential for continual learning applications.
- Extensive experiments have been performed to demonstrate the effectiveness of MMER on multi-modality expansion,  multi-modality retention and mitigating catastrophic forgetting.

**Weaknesses:**

- In terms of mitigating catastrophic forgetting, although MMER demonstrates good performance, considering the amount of storage required for MMER, it is unclear how this improvement compares to simply tuning task-specific adapters for new tasks.
- The paper mentioned that the hyper-parameters, such as $\alpha$ and $\lambda$, are selected based on the validation set. However, the construction of this validation set is not well-detailed, making it hard to tell how well these hyper-parameters can generalize to different tasks.
- The paper does not provide sufficient theoretical grounding for the proposed method, lacking analysis or reference on key aspects such as the relationship between model parameter differences and corresponding performance differences, the measurement of modality interference, or positive transfer between difference modality-specific LLMs.

**Questions:**

- In Equation 2, what is the rationale for using the Manhattan distance to minimize the distance between the reconstructed model parameters and the original LLM parameters?
- What will be the suggested way to quantify the difference between the original MLLM and its approximate version reconstructed by MMER? Furthermore, is there a relationship between this measured difference and the performance discrepancy?

---

> ### Author Response · Authors · 2024-11-20
> **Response to Reviewer Q9ea (1/3)**
>
> Thank you for your constructive comments. We will address your concerns point by point.
>
> > **Comment1**: In terms of mitigating catastrophic forgetting, although MMER demonstrates good performance, considering the amount of storage required for MMER, it is unclear how this improvement compares to simply tuning task-specific adapters for new tasks.
>
> **Reply**: We fine-tuned a LoRA adapter on original vision LLM for Flickr30k:
>
> |                 | VQAv2 |  GQA  | TextVQA | VizWiz | ScienceQA | POPE  | OK-VQA | 7 Image Tasks Avg | Flickr30k |
> | --------------- | :---: | :---: | :-----: | :----: | :-------: | :---: | :----: | :---------------: | :-------: |
> | Original MLLMs  | 78.11 | 61.52 |  55.89  | 51.51  |   71.12   | 86.17 | 31.33  |       62.23       |   51.26   |
> | Fine-tune       | 72.27 | 54.19 |  46.10  | 52.88  |   70.22   | 76.28 | 28.31  |       57.25       |   57.71   |
> | LoRA            | 75.72 | 58.24 |  52.87  | 52.64  |   70.63   | 85.08 | 29.21  |       60.63       |   54.85   |
> | **MMER (ours)** | 77.75 | 61.43 |  55.41  | 52.72  |   71.75   | 85.72 | 31.07  |       62.27       |   57.08   |
>
> The results show that LoRA improves the performance on target tasks but inevitably causes a decline on previous tasks, though the decline is less severe compared to full-parameter fine-tuning. **In contrast, our MMER approach outperforms LoRA on target tasks, while causing almost no degradation in previous tasks. However, the trade-off is a certain storage overhead.** Both approaches have distinct advantages and disadvantages, allowing users to choose based on their specific needs.
>
> More importantly, our approach has additional application scenario. In the open-source community, models are typically categorized into adapter-based models and full-parameter fine-tuned models. While the former can be easily integrated into existing models, the latter lacks such adaptability. **Our approach bridges this gap, providing a solution to seamlessly incorporate full-parameter fine-tuned models.**
>
>
>
> > **Comment2**: The paper mentioned that the hyper-parameters, such as α and λ, are selected based on the validation set. However, the construction of this validation set is not well-detailed, making it hard to tell how well these hyper-parameters can generalize to different tasks.
>
> **Reply**: We selected one task from each modality and combined their validation sets to construct this validation set. These tasks include **Objaverse**, **VocalSound**, **MSRVTT**, and **TextVQA**. These validation sets are all provided by the original dataset creators. The values of α and λ are selected based on the general performance of the merged MLLM on these validation sets. Re-tuning these hyperparameters for different tasks could lead to better results. Moreover, this calibration step is not complex and does not necessarily require dedicated validation sets. Instead, a small subset of samples from the training data is sufficient for calibration.
>
>
>
> > **Comment3**: The paper lacking analysis on the measurement of modality interference.
>
> **Reply**:  A straightforward measurement of parameter interference is by directly comparing performance. The more severe the parameter interference, the worse the performance.
> Besides, Ties [4] categorizes parameters by their importance and then evaluates interference based on the average values within each category. For instance, the lower the average value of irrelevant parameters, the less interference occurs.
>
> Lastly, our approach also preliminarily explores how to measure parameter interference. If the directions of the parameters are different, interference is clearly present. If the directions are the same, the greater the difference between the parameters, the more severe the interference. This is closely related to our approach, which minimizes the Manhattan distance between the reconstructed model parameters and the original LLM parameters. In other words, the difference between the original and merged parameters can serve as a measure of parameter interference. The closer the decoupled or merged parameters are to the original parameters, the less interference there is. Therefore, the Manhattan distance can be used to quantify parameter differences and, in turn, parameter interference.
>
> We measured the average Manhattan distance between the task vectors generated by three methods (**NaiveMC**, **Ties**, **MMER**) and the task vectors of four original MLLMs:
>
> - **NaiveMC**: 0.0056
> - **Ties**: 0.0008
> - **MMER**: 0.0002
>
> It can be observed that both Ties and MMER effectively reduce parameter interference to some extent. Among them, MMER achieves the smallest Manhattan distance, as it explicitly minimizes the Manhattan distance between the reconstructed model parameters and the original LLM parameters when constructing the task vector, thus mitigating parameter interference.

---

> ### Author Response · Authors · 2024-11-20
> **Response to Reviewer Q9ea (2/3)**
>
> > **Comment4**: The paper lacking analysis or reference on key aspects such as positive transfer between difference modality-specific LLMs.
>
> **Reply**: Firstly, we supplemented the MMER results on the new AVQA dataset, as well as the AVI-T modality results on the MUSCI-AVQA dataset.
>
> |                 | MUSCI-AVQA | MUSCI-AVQA | MUSCI-AVQA | MUSCI-AVQA | AVQA  | AVQA  | AVQA  | AVQA  |
> | --------------- | :--------: | :--------: | :--------: | :--------: | :---: | :---: | :---: | :---: |
> |                 |    IA-T    |    VI-T    |    VA-T    |   AVI-T    | IA-T  | VI-T  | VA-T  | AVI-T |
> | **MMER (ours)** |   47.25    |   51.27    |   51.77    |   53.54    | 78.13 | 79.74 | 79.98 | 80.72 |
>
> It can be observed that performance improves as the number of modalities increases when handling the same task. The introduction of additional modality information facilitates information sharing and knowledge complementarity between different modalities, thereby enabling positive transfer.
>
> Secondly, we observed overlaps between decoupled modality parameters and those of other modalities. Therefore, we removed these overlapping parameters to eliminate mutual interactions between modality parameters (i.e., changing the decoupling method from $m_i \circ \tau_*$ to $m_i \circ \tau_i$) and repeated both the multi-modality expansion and retention experiments. The results show that parameter overlaps slightly degraded performance, indicating that interaction between the overlapping parameters may have caused negative transfer.
>
> |                               | 3 Expansion Tasks |  11 Retention Tasks  |
> | ----------------------------- | :---------------: | :------------------: |
> |                               |     Avg ACC.      | Avg Score / Avg ACC. |
> | **MMER ($m_i \circ \tau_*$)** |       56.82       |    24.17 / 50.84     |
> | **MMER ($m_i \circ \tau_i$)** |       57.21       |    24.44 / 51.24     |
>
> In summary, from the perspective of input information, knowledge sharing and complementarity between modalities promote positive transfer. Whereas from the perspective of parameters, the overlap of modality parameters leads to negative transfer.
>
>
>
> > **Question1**: What will be the suggested way to quantify the difference between the original MLLM and its approximate version reconstructed by MMER? Furthermore, is there a relationship between this measured difference and the performance discrepancy?
>
> **Reply**: Model parameter differences can be quantified using various metrics, such as L1 or L2 distances, cosine similarity, or comparisons of hidden layer activations between models. However, these metrics do not necessarily correlate with performance discrepancy due to several key reasons:
>
> First, **parameter difference metrics are low-dimensional summaries, whereas model performance reflects high-dimensional behaviors**. Performance arises from complex interactions within the entire parameter space and its sensitivity to input data, which simple parameter metrics cannot fully capture.
>
> Second, **performance discrepancy depends on the specific task or data distribution**. Identical parameter difference may result in varying performance outcomes across different tasks, making it difficult to predict performance discrepancy without considering task-specific contexts.
>
> Additionally, many factors may influence the relationship between parameter differences and performance discrepancy, such as redundant parameters. DARE [3] points out that **there is a large amount of redundant parameters in the task vector, which complicates this task**. In this paper, the parameter differences between different models are quantified by calculating the cosine similarity of the embeddings generated for the same dataset. The results show that as more parameters are dropped, the cosine similarity of the embeddings continues to decrease, but the model performance remains almost unchanged initially, only collapsing towards the end.
>
> This is because initially dropped parameters are likely redundant, allowing performance to remain stable. It is only when key parameters are dropped that performance begins to degrade. Currently, we only know that low-magnitude parameters in the task vector are likely to be redundant, but we have not yet found a clear method to identify these redundant parameters. Therefore, the presence of redundant parameters makes it challenging to establish a direct relationship between parameter differences and performance discrepancy.

---

> ### Author Response · Authors · 2024-11-20
> **Response to Reviewer Q9ea (3/3)**
>
> > **Question2**: In Equation 2, what is the rationale for using the Manhattan distance to minimize the distance between the reconstructed model parameters and the original LLM parameters?
>
> **Reply**: First, for the sake of saving computational resources and improving efficiency, we did not adopt methods like Fisher [1] or Regmean [2], which require additional gradient-based computations to obtain the Information Matrix, as these methods demand substantial computational resources or data.
>
> Inspired by TIES [3] and DARE [4], which propose that “Supervised fine-tuned language models tend to acquire excessively redundant delta parameters”, we aim to decouple the most critical parameter of each modality from the merged task vector, so that the decoupled parameters are as close as possible to the original modality task vector.
>
> Considering the need to save storage and reduce computational resources, and based on DARE's [4] findings that key parameters are highly sparse, we decided to use a binary mask matrix to directly mask out irrelevant parameters in the merged task vector and only retain the key information related to the modality.
>
> We chose to use the Manhattan distance to optimize the mask mainly due to its mathematical properties and its promotion of sparsity in high-dimensional parameter spaces. Here are some detailed reasons and explanations:
>
> First, **Manhattan distance naturally facilitates parameter sparsification** because it tends to drive parameters to zero, which aligns perfectly with the binary mask matrix we aim to construct. The goal of the mask is to select key parameters relevant to a specific modality from the merged task vector and mask out irrelevant ones, meaning that most elements in the mask should be zero, with only a few elements set to 1. By minimizing the Manhattan distance, we can easily achieve this goal because the gradient of parameter updates with respect to Manhattan distance is constant. This makes it more likely to penalize smaller non-zero parameters and drive them to zero, thus encouraging the sparsity of the mask. Moreover, these smaller non-zero parameters are often redundant [4], which are the ones we wish to mask out.
>
> In contrast, Euclidean distance squares each difference, which results in larger magnitude parameter changes being penalized more significantly. **Therefore, it tends to shrink larger magnitude parameters, rather than smaller non-zero ones, which makes the mask tend towards smoothness rather than sparsity.** However, larger magnitude parameters are typically key and modality-specific [4], and this smoothing characteristic does not align with our need to construct the mask matrix. This smoothing effect could lead to the loss of critical parameter information. Hence, using Manhattan distance more effectively promotes the sparsity of the mask, extracting a modality-specific parameter subset.
>
> Furthermore, **Manhattan distance directly measures the element-wise difference between the merged task vector and the original modality task vectors**. This element-wise comparison can precisely capture which parameters have undergone significant changes during fine-tuning and which parameters are irrelevant noise. In contrast, **Euclidean distance emphasizes the “total distance” in the overall parameter space and fails to fully reflect the contribution of individual parameters**, making it less direct and effective than Manhattan distance in constructing the mask matrix.
>
> We have also conducted both the multimodal expansion and retention experiments that validate the effectiveness of Manhattan distance over Euclidean distance:
>
> |                               | 3 Expansion Tasks |  11 Retention Tasks  |
> | ----------------------------- | :---------------: | :------------------: |
> |                               |     Avg ACC.      | Avg Score / Avg ACC. |
> | **MMER (Manhattan distance)** |       56.82       |    24.17 / 50.84     |
> | **MMER (Euclidean distance)** |       56.05       |    23.89 / 50.41     |
>
>
>
> **Reference**
>
> [1] Matena et al. Merging Models with Fisher-Weighted Averaging. NeurIPS, 2022.
>
> [2] Jin et al. Dataless knowledge fusion by merging weights of language models. ICLR, 2023.
>
> [3] Yu et al. Language Models are Super Mario: Absorbing Abilities from Homologous Models as a Free Lunch. ICML, 2024.
>
> [4] Yadav et al. Ties-merging: Resolving interference when merging models. NeurIPS, 2023.

---

> > ### Comment · Reviewer_Q9ea · 2024-11-26
> >
> > Thank you for the response, which has addressed some of my concerns. I would like to maintain my current score of 6.

---

> > > ### Author Response · Authors · 2024-12-01
> > > **Replying to Reviewer Q9ea**
> > >
> > > Thank you for your positive score and insightful feedback. If you have any concerns about our work, we would greatly appreciate receiving any further comments or suggestions.

---

### Official Review · Reviewer_PSfy · 2024-11-04

**Soundness:** 3
**Presentation:** 3
**Contribution:** 3
**Rating:** 5
**Confidence:** 3

**Summary:**

The paper proposes a training-free approach, MMER, to enhance the multimodal capabilities of LLMs without extensive fine-tuning. To process different modalities, MMER maintains the vision encoder and introduces binary masks on the LLM parameters for each modality, facilitating modality-specific input processing and reducing parameter conflicts. Additionally, MMER incorporates a similar method to mitigate catastrophic forgetting by decoupling parameters fine-tuned on new tasks from the original parameters. The experimental results on multimodal and dual-modal tasks indicate that MMER achieves notable improvements over recent baselines, demonstrating its potential effectiveness.

**Strengths:**

- MMER’s training-free approach makes it practical and resource-efficient, avoiding the computational costs of extensive fine-tuning.
- The method effectively retains the original performance of merged models, mitigating catastrophic forgetting and preserving their capabilities.
- MMER demonstrates versatility, applicable to various modalities, and maintains performance across diverse multimodal tasks.
- The paper includes extensive experiments that show consistent performance improvements over baseline methods, validating MMER’s robustness.

**Weaknesses:**

- The paper could better explain the advantages of MMER over existing modular approaches and provide a clearer justification for adopting this monolithic method.
- While the paper demonstrates effectiveness, it lacks comparisons with certain mainstream MLLMs and does not evaluate larger-scale models. Including these aspects would strengthen the argument for MMER’s superiority and provide insights into its performance at scale.

**Questions:**

Refer to the weakness.

---

> ### Author Response · Authors · 2024-11-20
> **Response to Reviewer PSfy**
>
> Thank you for your valuable comments. We will address your concerns point by point.
>
>
>
> > **Comment1**: The paper could better explain the advantages of MMER over existing modular approaches and provide a clearer justification for adopting this monolithic method.
>
> **Reply**: We have provided a detailed discussion of the advantages and novelty of MMER over existing modular approaches in **Appendix A**, which you can refer to. Below, we summarize these points and offer a clearer justification for adopting this monolithic method:
>
> - **Enhanced Multimodal Performance**:
>
>   Compared to other training-free model merging methods, our MMER approach effectively enhances the multimodal performance of the merged MLLM.
>
> - **Retention of Original Performance**:
>
>   MMER can retain the original model's performance while simultaneously enhancing multimodal capabilities, a challenge that other model merging methods often struggle to overcome.
>
> - **Mitigating Catastrophic Forgetting**:
>
>   MMER additionally alleviates catastrophic forgetting, an aspect that has not been explored by other model merging  methods.
>
> - **Training-Free**:
>
>   MMER offers a training-free solution for multimodal expansion, unlike fine-tuning methods that require large datasets and substantial computational resources.
>
>
>
>
>
> > **Comment2**: While the paper demonstrates effectiveness, it lacks comparisons with certain mainstream MLLMs.
>
> **Reply**: We have supplemented the results of mainstream MLLMs, such as ImageBind-LLM, OneLLM-7B, and X-InstructBLIP, on multimodal benchmarks and observed that MMER achieves superior performance compared to them.
>
> | Task (→)        | ModelNet40 | MUSCI-AVQA | MUSCI-AVQA | MUSCI-AVQA | MCUB  | MCUB  | MCUB  | MCUB  |  MCUB  |
> | --------------- | :--------: | :--------: | :--------: | :--------: | :---: | :---: | :---: | :---: | :----: |
> | **Model (↓)**   |    PI-T    |    IA-T    |    VI-T    |    VA-T    | AVI-T | AVP-T | AIP-T | VIP-T | AVIP-T |
> | OneLLM-7B       |     -      |   44.64    |   45.48    |   47.60    |   -   |   -   |   -   |   -   |   -    |
> | ImageBind-LLM   |   39.86    |   36.54    |   38.76    |   39.72    | 35.20 | 31.40 | 33.40 | 31.80 | 32.93  |
> | X-InstructBLIP  |   57.93    |   40.71    |   41.23    |   48.34    | 41.40 | 25.20 | 21.20 | 29.40 | 27.94  |
> | **MMER (ours)** |   62.15    |   47.25    |   51.27    |   51.77    | 56.48 | 59.31 | 65.59 | 56.00 | 61.63  |
>
>
>
>
>
>
>
> > **Comment3**: While the paper demonstrates effectiveness, it does not evaluate larger-scale models.
>
> **Reply**: We understand the importance of evaluating larger-scale models. However, we would like to clarify the following points regarding the current scope of our work:
>
> 1. In the open-source community, the most commonly available MLLMs are those with 7B parameters. Finding four distinct MLLMs that share both the same architecture and parameter size exceeding 7B, particularly across multiple modalities, is extremely challenging at present. Therefore, conducting experiments on models larger than 7B is currently not feasible due to the limited availability of such models.
>
> 2. Previous experiments on model merging and multimodal expansion have also focused on models with a maximum of 7B parameter. There has been little exploration of these methods on models with significantly larger-scale parameter sizes, as such models are not yet commonly available in the field.
>
> 3. **Experiments on larger-scale models do not affect the contribution of our work.** For example, earlier model merging methods such as TIES [1] and Task Arithmetic [2] were initially explored on smaller models like T5, BERT, or ViT, but subsequent research has shown that they are equally effective for 7B LLMs. Therefore, I believe that the model size also does not pose an obstacle for our MMER approach.
>
> **Reference**
>
> [1] Yadav et al. Ties-merging: Resolving interference when merging models. NeurIPS, 2023.
>
> [2] Ilharco et al. Editing models with task arithmetic. ICLR, 2023.

---

### Official Review · Reviewer_VHBG · 2024-11-04

**Soundness:** 3
**Presentation:** 4
**Contribution:** 3
**Rating:** 6
**Confidence:** 4

**Summary:**

This paper introduces a training-free parameter-merge framework, called MMER, for MLLMs. This method compresses key parameter differences between each MLLM and original LLM into a merged task vector and a multimodal mask. When addressing tasks involving multimodal expansion and retention, MMER restores the parameters specific to the required modality using the task vector and mask, making the merged model approximate the corresponding MLLM. Extensive experiments demonstrate MMER’s effectiveness across various benchmarks and its capability to mitigate catastrophic forgetting.

**Strengths:**

1. The paper offers several clear illustrations that is greatly helpful in understanding the mechanisms of parameter merging, expansion, and retention.
2. By decoupling the merged task vector and mask, the model can utilize parameters approximating to specific modality to process corresponding modality inputs, which sounds make sense.
3. This method merges the newly fine-tuned MLLM as an additional task vector, while maintaining performance on the original task.

**Weaknesses:**

For audio modality, the merged model only selects 2.2% parameters from merged task vector (Figure 4a), which indicates the majority of key parameters in the audio MLLM deviate from the direction of the original LLM according to the merging mechanism. However, the merged model (98% parameter from original LLM + 2% parameter from merged task vector activated by audio mask) achieves 1.5x improvement compared to original audio MLLMs, which is confused.

The author attributes this result to the redundant parameters in the audio model, consistent with the observations in [1]. However, even in [1], as more parameters are dropped, performance declines rather than an exaggerated 1.5x improvement. Considering that 2% of the parameters are not from the original audio MLLM, I wonder if that 2% of the parameters overlap with other modalities (such as video), thus providing more prior knowledge?

[1] Yu et al. Language Models are Super Mario: Absorbing Abilities from Homologous Models as a Free Lunch.

**Questions:**

According to the merging mechanism, merged task vector retains the parameters of target MLLM that are consistent with those of original LLM in direction. As more modalities are merged, the merged task vector will also accumulate key parameters from other modalities. Will the parameters of a single modality correlate to the final merged task vector? Will this correlation weaken with the introduction of more MLLMs, leading to the failure of the merging mechanism? In the merged task vector, what is the parameter overlap like across different modalities?

---

> ### Author Response · Authors · 2024-11-20
> **Response to Reviewer VHBG (1/2)**
>
> Thanks for your constructive comments. We will address your concerns point by point.
>
>
>
> > **Comment1**: For audio modality, the merged model only selects 2.2% parameters from merged task vector (Figure 4a), which indicates the majority of key parameters in the audio MLLM deviate from the direction of the original LLM according to the merging mechanism. However, the merged model achieves 1.5x improvement compared to original audio MLLMs, which is confused.
> >
> > The author attributes this result to the redundant parameters in the audio model, consistent with the observations in [1]. However, even in [1], as more parameters are dropped, performance declines rather than an exaggerated 1.5x improvement. Considering that 2% of the parameters are not from the original audio MLLM, I wonder if that 2% of the parameters overlap with other modalities (such as video), thus providing more prior knowledge?
>
> **Reply**: You may have misunderstood our approach. Let me clarify. We construct the mask by comparing the merged task vector with the original MLLM task vector. Therefore, the selection of only 2.2% of parameters by the audio mask does not mean most key parameters in the audio MLLM deviate from the direction of the original LLM. Instead, it reflects the difference between the merged task vector and the original audio MLLM task vector.
>
> Moreover, parameters not being selected does not only mean that their directions deviate. There are two possible cases for parameters not being selected:
>
> 1. The directions are opposite.
> 2. The directions are the same, but the magnitude of the parameters in the original audio MLLM task vector are too small.
>
> To further clarify this, we examined the percentage of parameters in the merged task vector whose directions are consistent with those in the task vectors of the four modalities:
>
> - Audio (A): 50.62%
> - Video (V): 57.58%
> - Image (I): 69.20%
> - Point (P): 70.09%
>
> It can be observed that 49.38% of audio parameters are not selected due to opposite directions, while the remaining 48.42% are not selected because their magnitude are relatively small. Therefore, we examined the average magnitude of the task vectors for the four modalities:
>
> - Audio (A): 8.4e-5
> - Video (V): 2e-4
> - Image (I): 5e-4
> - Point (P): 5e-4
>
> We found that the magnitude of the audio task vector are significantly smaller than those of the other modalities, which explains why the audio mask only selected 2.2% of the parameters. This indicates that **the original audio MLLM is highly similar to the pre-trained LLM**. As a result, the merged model (98% of the parameters from the pre-trained LLM + 2% of the parameters activated by the audio mask from the merged task vector) only needs to activate 2.2% of the key parameters to retain its performance.
>
> **As for why a 1.5x improvement can be achieved, we provide the following explanation:**
>
> The performance improvement is not due to the removal of redundant parameters. Generally, as more parameters are dropped, performance tends to decline [1,2]. This trend was also observed in our analysis **(see Figure 4b)**, where increasing the Dominant Significance λ·50% led to fewer parameters being selected for each modality, resulting in a gradual performance drop.
>
> So why does it achieve a 1.5x improvement? Your hypothesis is correct: the 2.2% of parameters selected have overlaps with parameters from other modalities. To investigate this, we examined the overlap of these 2.2% parameters with those of other modalities: **41.7% of these parameters do not overlap with any other modality, while 23.2%, 21.1%, and 22.1% overlap with the video, image, and point modalities, respectively.**
>
> The model may benefit from some additional knowledge contained in these overlapping parameters, such as prior knowledge or instruction-following capabilities. We have analyzed this phenomenon in **lines 413–420** of the paper. To verify this, **we replaced the overlapping parameters with the original audio task vector's parameters and conducted experiments on three audio tasks, yielding results of 24.71 (97.6) / 24.32 (98.4)**. It can be observed that the 1.5x improvement in performance was lost, confirming the validity of our analysis.

---

> ### Author Response · Authors · 2024-11-20
> **Response to Reviewer VHBG (2/2)**
>
> > **Question1**: According to the merging mechanism, merged task vector retains the parameters of target MLLM that are consistent with those of original LLM in direction. As more modalities are merged, the merged task vector will also accumulate key parameters from other modalities. Will the parameters of a single modality correlate to the final merged task vector? Will this correlation weaken with the introduction of more MLLMs, leading to the failure of the merging mechanism? In the merged task vector, what is the parameter overlap like across different modalities?
>
> **Reply**: Yes, the parameters of a single modality are correlated with the final merged task vector. However, this correlation weakens as more models are introduced. We also observed this trend in our analysis **(see Figure 5a)**: regardless of the merging method used, the more models are merged, less information from each individual model is retained, leading to performance degradation. This phenomenon was also observed in Ties [2], which merged up to seven models. This increasing parameter interference ultimately causes the merging mechanism to fail. This issue is an inherent limitation of model merging—no algorithm can perfectly accommodate the merging of an unlimited number of models. We can only optimize algorithms to mitigate parameter interference and minimize the extent of performance degradation. In **Figure 5a**, it can be seen that MMER experiences only minor performance degradation compared to other merging methods, demonstrating the effectiveness of MMER in decoupling parameters.
>
> We analyzed the overlap of parameters from different modalities within the merged task vector:
>
> - Specifically, 40.43% of audio parameters, 55.36% of video parameters, 64.49% of image parameters, and 66.28% of point parameters are integrated into the merged task vector.
> - The percentage of parameters unique to each modality in the merged task vector is as follows:
>   - Audio (A): 1.16%
>   - Video (V): 2.41%
>   - Image (I): 6.27%
>   - Point (P): 8.24%
> - The percentage of overlap between parameters from two modalities is as follows:
>   - A & V: 22.38%
>   - A & I: 25.03%
>   - A & P: 25.77%
>   - V & I: 34.93%
>   - V & P: 33.33%
>   - I & P: 36.50%
>
> The analysis reveals severe overlap between parameters from different modalities. This highlights the necessity of MMER in decoupling key parameters to mitigate parameter conflicts effectively.
>
>
>
> **Reference**
>
> [1] Yu et al. Language Models are Super Mario: Absorbing Abilities from Homologous Models as a Free Lunch. ICML, 2024.
>
> [2] Yadav et al. Ties-merging: Resolving interference when merging models. NeurIPS, 2023.

---

> > ### Comment · Reviewer_VHBG · 2024-11-29
> >
> > Thanks for your effort in rebuttal. Please update the paper with these feedback. I think the rating should be 7 (but there is no such score), so I keep the original score.

---

> > > ### Author Response · Authors · 2024-12-01
> > > **Replying to Reviewer VHBG**
> > >
> > > Thank you for your positive score and insightful feedback. Unfortunately, the deadline for revising the paper has passed by the time of your response, but we had already made updates to certain parts of the paper before that. The remaining content will be further revised in the final version (if the paper is accepted). If you have any concerns or suggestions about our work, we would greatly appreciate receiving any further comments or suggestions.

---

### Author Response · Authors · 2024-11-20
**General Response**

We appreciate your consideration in taking the time to review our comments. We received valuable feedback from four reviewers, who provided insightful and constructive comments. Below, we summarize the key strengths highlighted in their feedback:

1. **Clarity and Presentation**:

   The paper is well-organized, clearly articulates the problem, and presents the proposed solution effectively. The use of clear illustrations and tables greatly aids in understanding the mechanisms of parameter merging, expansion, and retention, making the paper accessible and easy to follow.

2. **Innovation and Practicality**:

   The training-free MMER approach offers a practical, resource-efficient solution for multi-modality expansion in LLMs, eliminating the need for costly extensive fine-tuning. The novel parameter decoupling strategy enables the separation of modality-specific parameters, reducing interference and enhancing performance.

3. **Effectiveness and Versatility**:

   MMER shows versatility by supporting multi-modality expansion, retaining original performance, and mitigating catastrophic forgetting across diverse modalities.

4. **Comprehensive Experiments**:

   Extensive experiments validate MMER’s effectiveness, showcasing consistent improvements over baseline methods. The results underline MMER’s robustness in multi-modality expansion, retention, and continual learning applications.

However, there are still some questions and concerns from the reviewers. We have summarized and addressed these in four key points:

1. **Principles of Model Merging and Decoupling**:

   The principles and mechanisms of model merging and decoupling based on task vectors have been explored and proven in previous studies [1,2,3,4]. Building on this foundation, we provide a comprehensive summary and analysis. Specifically, we analyze the merging mechanism from the perspective of the number of merged models (**see Response to Reviewer VHBG, Question 1**) and explain why arithmetic operations can be applied to task vectors to achieve model merging and decoupling (**see Response to Reviewer egSN, Comment 3**).

2. **Rationale for Using Manhattan Distance**:

   The reasons for using the Manhattan distance for calculating the mask are threefold. First, to save computational resources and improve efficiency, we opted for a binary mask that directly eliminates irrelevant parameters in the merged task vector, retaining only the key information related to the modality. Second, the Manhattan distance naturally encourages parameter sparsification, which aligns well with the binary mask, whereas the Euclidean distance makes the mask tend towards smoothness. Finally, the Manhattan distance measures element-wise differences, whereas the Euclidean distance emphasizes the "total distance" in the entire parameter space. And experiments are conducted to validate this analysis (**see Response to Reviewer egSN, Comment 4**).

3. **Experimental Supplements:**

   We have additionally included comparison experiments with LoRA on mitigating catastrophic forgetting (**see Response to Reviewer Q9ea, Comment 1**), comparisons with mainstream MLLMs (**see Response to Reviewer PSfy, Comment 2**), the results of four original unimodal models on tri-modality tasks (**see Response to Reviewer egSN, Comment 2**), and results from new cross-modal tasks (**see Response to Reviewer egSN, Comment 1**). The results of these experiments further strengthen the validation of the effectiveness of our MMER method.

4. **Analysis of Methods and Experiments:**

   We have supplemented additional analyses on the method and experiments to validate the rationality of our approach while providing theoretical support. Specifically, we analyzed the overlap in decoupled parameters (**see Response to Reviewer VHBG, Comment 1 & Question1**) and examined the measurement of modality interference (**see Response to Reviewer Q9ea, Comment 3**). Additionally, we explored the positive transfer between different modality-specific LLMs (**see Response to Reviewer Q9ea, Comment 4**) and investigated the relationship between parameter differences and performance discrepancy (**see Response to Reviewer Q9ea, Question 1**).

Once again, we sincerely thank you for your involvement and thoughtful feedback! We have provided detailed responses to each question from each reviewer.

**Reference**

[1] Ortiz-Jimenez et al. Task Arithmetic in the Tangent Space: Improved Editing of Pre-Trained Models. NeurIPS, 2023.

[2] Ilharco et al. Editing models with task arithmetic. ICLR, 2023.

[3] Yu et al. Language Models are Super Mario: Absorbing Abilities from Homologous Models as a Free Lunch. ICML, 2024.

[4] Yadav et al. Ties-merging: Resolving interference when merging models. NeurIPS, 2023.

---

### Meta-Review · Area_Chair_eA2y · 2024-12-15

**Metareview:**

This paper introduces MMER, a training-free approach for multimodal expansion and retention in large language models (LLMs), leveraging parameter merging and decoupling. Strengths include its resource-efficient design, robustness in mitigating catastrophic forgetting, and extensive experiments demonstrating competitive performance across multimodal tasks. However, reviewers noted limitations in theoretical grounding, insufficient diversity in task evaluations, and limited clarity on certain methodological choices. While the authors provided detailed rebuttals and addressed several concerns, the revisions only partially resolved key issues.

**Additional Comments On Reviewer Discussion:**

Reviewers raised questions about the clarity and theoretical basis of MMER, its task coverage, and hyperparameter selection. The authors clarified the rationale for their design, supplemented experiments, and expanded comparisons with baseline methods. Despite these efforts, some concerns, such as the generalization of hyperparameters and a lack of broader task validation, remained partially addressed, leading reviewers to maintain their initial scores.

---

### Decision · Program_Chairs · 2025-01-22

Reject